. Pathogens

# A comprehensive PDCoV-host proteome interaction map reveals potential antiviral targets

Wenjun Yan[1☉], Kailu Wang[1☉], Song Liu[1☉], Rongbin Qiu[2☉], Qingcheng Yang[1], Hao Li[1], Siyu Huang[1], Chengyao Hou[1], Qinyuan Chu[1], Yue Sun[1], Yizhi Tang[1], Changwei Lei[1], Yiming Tian[3], Hongning Wang[1]*, Xin Yang[1]*

**1** Animal Disease Prevention and Green Development Key Laboratory of Sichuan Province, College of Life Science, Sichuan University, Chengdu, China, **2** College of Life Science and Engineering, Southwest University of Science and Technology, Mianyang, China, **3** Guizhou Medical University, Guiyang, China

☉ These authors contributed equally to this work.
* whongning@163.com (HW); yangxin0822@163.com (XY)

## Abstract

Porcine deltacoronavirus (PDCoV), an enteric member of the coronavirus family, has emerged globally over the past decade, causing significant impacts on the swine industry. While studies of virus-host protein interactions provide crucial insights into viral engagement with host cells during infection, research specifically targeting PDCoV-host interaction factors remains limited. To identify host proteins involved in PDCoV replication, comprehensive identification of RNA-binding proteins by mass spectrometry (ChIRP-MS) was employed to identify host proteins interacting with the PDCoV genomic RNA. Concurrently, affinity purification mass spectrometry (AP-MS) was utilized to identify host interactors of PDCoV-encoded proteins. A total of 671 host proteins were identified in our analysis. These host interactors participate in diverse cellular processes, including extensive representation of metabolic enzymes, transcription factors, RNA-binding proteins (RBPs), and intracellular signal transduction components. Construction of a comprehensive PDCoV-host protein interaction network map revealed that SYNCRIP (heterogeneous nuclear ribonucleoprotein Q, hnRNP Q), functions as a novel host restriction factor with PDCoV. SYNCRIP interacts with the N proteins of multiple coronaviruses and competitively displaces HUWE1 to bind the PDCoV N protein, thereby blocking its ubiquitin-proteasome-mediated degradation. Furthermore, Isoforsythiaside, a small-molecule inhibitor designed to target SYNCRIP, demonstrated substantial antiviral potential both in vitro and in vivo. In summary, this study provides a comprehensive catalog of functional PDCoV viral RNA (vRNA)/viral Protein (vProtein)-host protein interactions. This resource not only informs the understanding of pan-coronavirus infection mechanisms but also nominates host cellular processes as potential targets for antiviral intervention.

**Data availability statement:** All relevant data are in the manuscript and its supporting information files.

**Funding:** This work was supported by the National Natural Science Foundation of China (Grant No. 32260035 to X. Y.), and the Major Science and Technology Special Projects of Sichuan Province (Grant No. 2021ZDZX0010 to X. Y.). The funders had no role in study design, data collection and analysis, decision to publish, or preparation of the manuscript.

**Competing interests:** The authors have declared that no competing interests exist.

## Author summary

Porcine deltacoronavirus (PDCoV) demonstrates significant cross-species infection potential, with its spillover risk to human hosts posing a major public health threat. Elucidating the molecular interaction mechanisms between the virus and host is of critical scientific value for antiviral drug target discovery. In this study, we systematically constructed a virus-host interaction network centered on PDCoV's infectious RNA and structural proteins, identifying host interaction proteins involved in key biological processes, including metabolic regulation, translation and transcription, vesicular transport, and ubiquitination modification. Network analysis revealed the molecular mechanism by which PDCoV hijacks host cells for life program reprogramming. Further functional investigation of the key hub protein SYNCRIP demonstrated that it significantly promotes viral replication by competitively binding to the viral N protein, thereby blocking the ubiquitination and degradation pathway mediated by HUWE1. In vivo and in vitro experiments confirmed that targeting SYNCRIP exerts potent antiviral effects, providing a novel strategy for antiviral drug development based on virus-host interaction networks. This study validates that systematic analysis of virus-host interaction networks offers significant advantages in identifying novel antiviral targets.

## Introduction

The global pandemic of severe acute respiratory syndrome coronavirus 2 (SARS-CoV-2), the etiological agent of coronavirus disease 2019 (COVID-19) with significant mortality rates, exemplifies the formidable challenges RNA viruses present to global health security [1,2]. Of particular concern is PDCoV, an emerging enteropathogenic coronavirus that causes severe, often lethal diarrhea in neonatal piglets. Its demonstrated capacity for cross-species transmission raises substantial zoonotic concerns, warranting urgent scientific attention [3,4]. PDCoV is an enveloped virus with a single-stranded RNA genome approximately 25.4 kb in length, containing nine open reading frames (ORFs). Exhibiting characteristic coronavirus features, the two-thirds of the PDCoV genome encodes two large polyproteins, pp1a and pp1ab. These precursors are subsequently cleaved by papain-like protease (PLpro) and 3C-like serine protease (3CLpro) to generate 15 mature nonstructural proteins (NSPs) essential for viral replication. The remaining one-third of the genome encodes four structural proteins—spike (S), envelope (E), membrane (M), and nucleocapsid (N)—along with three accessory proteins: NS6, NS7, and NS7a [5]. Viral infections exhibit an obligate dependence on host biomolecular interactions to complete their replicative cycle [6]. The infection process involves multiple intricate, virus-specific steps: cellular attachment and entry, followed by hijacking of host machinery. Upon internalization, viral genetic material systematically co-opts and remodels cellular pathways. Beyond vRNA, numerous vProteins form extensive interaction networks with host proteins - either directly or through intermediary complexes - to facilitate viral gene expression,

genome replication, and virion assembly [7]. Thus, comprehensive mapping of virus-host protein-protein interactions (PPIs) provides fundamental insights into infection mechanisms and host defense strategies, offering critical avenues for therapeutic development. Currently, multiple methods are available to assess the interactions between viral-encoded proteins and host proteins, including yeast two-hybrid (Y2H), BioID-MS, and AP-MS. Among these, AP-MS is the most widely used approach. Gordon *et al.* were the first to establish a SARS-CoV-2-host protein-protein interaction network using AP-MS, which identified 389 virus-host interacting proteins that provided significant value for early antiviral therapeutic target discovery [8]. Another study further expanded the SARS-CoV-2-associated host interactome by combining AP-MS with Y2H [9]. PPI networks assume particular significance as proteins rarely function in isolation but rather through coordinated complexes. Previous RNA-centric studies have elucidated host recognition mechanisms targeting viral genomes [2,10,11]. ChIRP-MS employs biotinylated probes that specifically bind to the target RNA to pull down RNA-associated proteins, followed by mass spectrometry identification. This approach enables the discovery of host proteins naturally bound to the target RNA. The method has been successfully applied to identify host proteins interacting with ZIKV, EBOV, and SARS-CoV-2 [12]. Complementarily, systematic PPI mapping has revealed functionally interconnected host pathways subverted during infection [8,13]. Such integrated approaches combining vRNA/vProtein-level analyses with proteomic interaction data - substantially advance our understanding of RNA virus pathogenesis while identifying multiple targetable vulnerabilities for antiviral intervention.

The HnRNPs (heterogeneous nuclear ribonucleoproteins) family constitutes a highly abundant class of RNA-binding proteins in eukaryotic cells, involved in RNA transcription, splicing, translation, and other processes. Among them, HnRNPQ (SYNCRIP) is a core member of the HnRNP family, containing multiple RNA recognition motif (RRM) domains. Previous studies have demonstrated that SYNCRIP binds to the NS1 mRNA of porcine parvovirus (PPV) and promotes the cleavage of NS1 mRNA into NS2 mRNA [14]. Antiviral therapeutics are broadly categorized into two classes: direct-acting antivirals (DAAs) that target viral components (e.g., inhibiting viral polymerases and proteases), and host-directed antivirals (HDAs) that disrupt virus-host PPI by targeting host proteins essential for viral replication. Notably, the accelerated evolutionary rate of viruses frequently leads to treatment failure due to drug resistance in DAA regimens—a phenomenon particularly pronounced during RNA virus infections [15,16]. In contrast, host proteins exhibit slower evolutionary rates, and multiple viruses often hijack identical host factors; for example, YBX1 recognizes diverse viral RNAs to facilitate replication [17–19]. Thus, comprehensive mapping of PDCoV-host protein interactions represents a strategic imperative for developing effective countermeasures against viral dissemination.

In this study, we employed ChIRP-MS to identify host factors interacting with the PDCoV genomic RNA. Complementarily, AP-MS was implemented to characterize the host protein interactome of PDCoV-encoded proteins. Through integration of these approaches, we established a global PDCoV-host protein interaction map, identifying 671 host proteins interacting with PDCoV components. Significantly, 102 proteins were shared between vRNA-host and vProtein-host interactomes, including factors involved in signal transduction, host restriction, vesicular trafficking, and nuclear RNA processing/export. Functional characterization of core interactors identified SYNCRIP as a bona fide host restriction factor helping both PDCoV and porcine epidemic diarrhea virus (PEDV) replication. Small-molecule inhibitors targeting SYNCRIP demonstrated potent therapeutic efficacy. Collectively, these findings highlight the translational potential of our comprehensive PDCoV-host interactome for discovering pan-antiviral therapeutic targets directed against host dependency factors.

## Results

### Integrated ChIRP-MS and AP-MS analysis identifies host interactors of PDCoV genomic RNA and encoded proteins

PDCoV, an emerging deltacoronavirus, poses significant threats to human health due to its broad host adaptability. While the "spillover" potential of PDCoV is recognized, the specific cellular pathways and host proteins involved in its infection

remain poorly understood [4]. To comprehensively characterize PDCoV-host interactions, we systematically identified host proteins interacting with vRNA and vProteins, establishing a PDCoV-host protein interaction network. We utilized the laboratory-maintained PDCoV strain SCCZ18 (NCBI accession: MT985156), featuring a 25.4 kb genome with typical coronavirus organization encoding 14 nonstructural proteins (nsp2-nsp16), 4 structural proteins, and 2 accessory proteins (Fig 1A). For vRNA-host protein identification, we performed chromatin isolation by ChIRP-MS. This formaldehyde crosslinking-based approach preserves native RNA-protein complexes and has been successfully applied to study host interactions of ZIKV, EBOV, and SARS-CoV-2 [12]. Considering PDCoV's cell tropism variations, we first evaluated viral replication kinetics in LLC-PK1 and ST cells. At MOI = 0.01, both cell lines exhibited mild cytopathic effects (CPE) at 24 hpi and pronounced CPE at 48 hpi, with viral replication confirmed by WB and RT-qPCR (S1A Fig). Infected cells (24/48 hpi) were formaldehyde-fixed for RNA-protein crosslinking. As direct vRNA labeling was impractical, we designed 108 biotinylated oligonucleotide probes (S1 Table) targeting the full-length positive-strand vRNA. The probe pool was incubated with infected cell lysates to form vRNA-probe-host protein complexes, which were then streptavidin-purified for LC-MS/MS analysis (Fig 1B, left). For vProtein-host interactions, we employed AP-MS. All 20 mature viral proteins were codon-optimized and cloned into mammalian expression vectors with N-terminal 2 × Strep tags. To maximize physiological relevance, we used PDCoV-susceptible porcine LLC-PK1 cells (despite lower transfection efficiency than 293T cells) for affinity pull-down experiments. Purified complexes were analyzed by LC-MS/MS (Fig 1B, right).

## Comprehensive identification of host factors interacting with PDCoV

**A comprehensive atlas of host-factors that interact with the PDCoV genomic RNA.** Protein silver staining of input and output samples from ChIRP revealed negligible staining in control samples, while a prominent band at ~40 kDa was observed in all samples from PDCoV-infected cells across both cell lines. This molecular weight corresponds to the viral N protein (Fig 2A). Technical quality of ChIRP was assessed by analyzing recovered viral and host RNAs. As expected, RT-qPCR confirmed substantial viral RNA recovery (S1B Fig). Given the unique replication mechanism of coronaviruses, which generates abundant genomic mRNA (gmRNA) and subgenomic mRNAs (sgmRNAs) during replication [20,21], we examined the gmRNA/sgmRNA ratio and observed higher sgmRNA levels in the recovered RNA (S1C Fig), consistent with prior studies on SARS-CoV-2 [2]. Correlation analysis across experimental replicates (n = 3) confirmed high reproducibility (S1D Fig). Collectively, these quality assessments at both protein and RNA levels demonstrate that ChIRP is effective for screening PDCoV RNA-host protein interactions. ChIRP-MS identified numerous host proteins, which were screened using FC > 2 and p < 0.05/0.01 as thresholds for significant/highly significant differences. Multiple viral proteins, including N, S, M, and ORF1ab, were reliably detected in both cell lines, with N protein binding viral RNA to form RNP complexes. A total of 256 (LLC-PK1) and 169 (ST) host factors interacted with vRNA (Fig 2B). The top 50 host factors by FC value in each cell line included YBX1, TRIM28, and G3BP1—proven critical in viral infection [22–24]. Analysis of vRNA-binding proteins at 24 hpi and 48 hpi revealed temporal differences in host factor recruitment: only 18.75% (48/256) were shared at both time points in PK1 cells, compared to 21.3% (36/169) in ST cells (Fig 2C). Host factors persistently associated across time points are listed (S1I). A core set of 67 host factors bound vRNA in both cell lines (Fig 2D), totaling 357 identified PDCoV RNA-interacting host factors (S2 Table).

Cytoscape visualized the vRNA-host protein interactome, with nodes representing significantly enriched proteins from ChIRP-MS data. Functional categorization highlighted diverse pathways: metabolic enzymes, translation machinery, cytoskeletal components, and abundant RBPs dominated in PK1 cells (S2A Fig), similarly enriched in ST cells (S2B Fig). Unexpected pathways included FAD-binding proteins, intracellular vesicle proteins, cytoplasmic signaling mediators, and numerous hnRNPs family members. Temporal comparisons (24 hpi vs. 48 hpi) showed ATP-binding proteins, protein metabolism regulators, nuclear complexes, and cyto-signaling enriched at 24 hpi in PK1, while hnRNPs peaked at 48 hpi—a pattern replicated in ST cells. Cross-cell-line analysis revealed conserved binding of core RBPs, whereas cell adhesion proteins, methylation regulators, and serine/arginine-rich (SR) proteins were PK1-specific, and poly-A-binding

# A

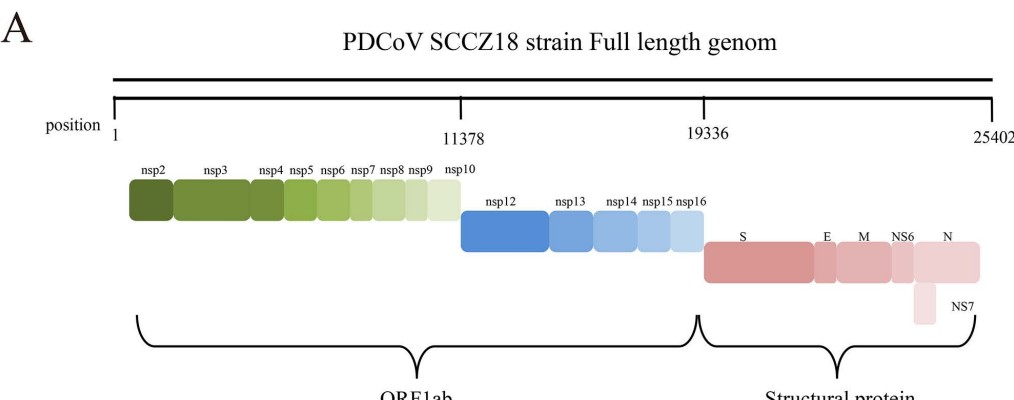

# B

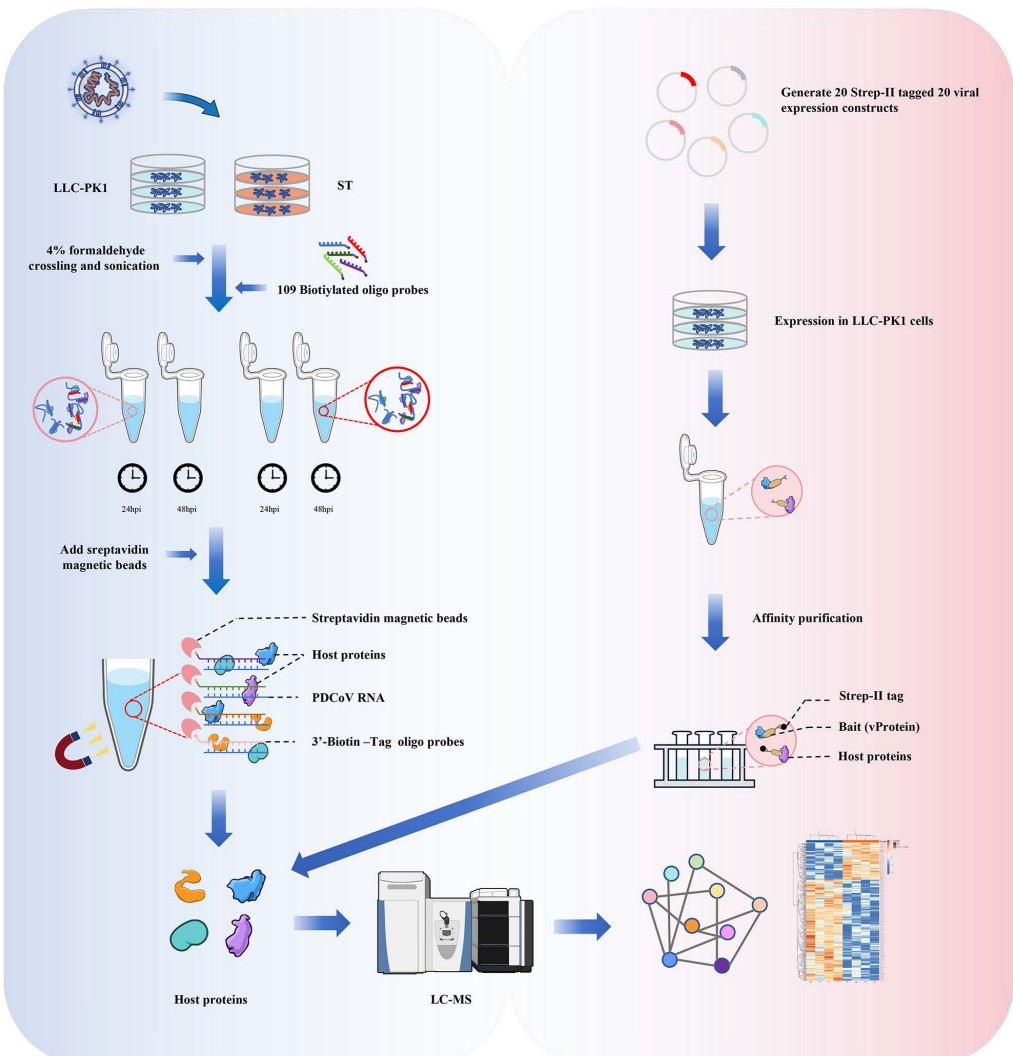

**Fig 1. (A) Genomic characteristics of the PDCoV SCCZ18 strain.** (B) Schematic workflow of ChIRP-MS (left) for identifying proteins binding to PDCoV vRNA, and AP-MS protocol (right) for detecting proteins interacting with PDCoV vProtein. Illustration from NIAID NIH BioArt Source (bioart.niaid.nih.gov/bioart/).

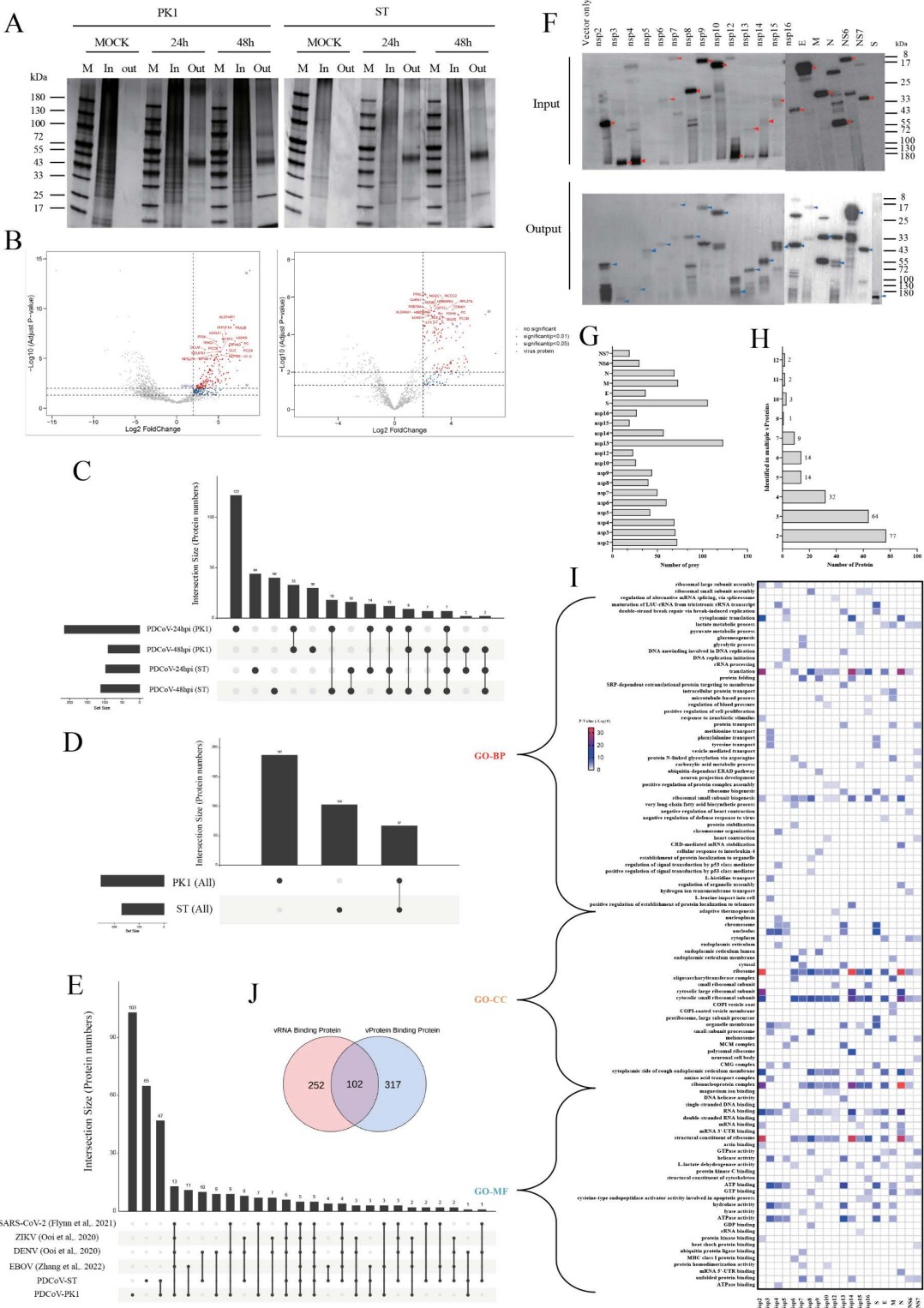

**Fig 2. (A) Silver-stained SDS-PAGE analysis of total protein input and output samples from mock-infected or PDCoV-infected (24/48hpi) cells using biotinylated oligonucleotides targeting PDCoV.** (B) ChIRP-MS data processed by R package "DEP" (FC > 2, p < 0.05 or p < 0.01 as significance thresholds), showing host binding proteins in PDCoV-infected LLC-PK1 (left) and ST cells (right). (C) Temporal dynamics of PDCoV-host protein

interactions across infection timepoints. (D) Cell line-specific differences in PDCoV-host protein interactions. (E) Overlap between PDCoV-interacting proteins and host factors of other RNA viruses. (F) WB verification of Strep II-tagged vProtein expression in PK1 cells and affinity-purified samples. (G) Host protein interaction sets for individual vProteins. (H) Distribution of host proteins across different vProtein interaction sets. (I) GO analysis of host proteins interacting with each vProtein (Top 5 terms shown per set). (J) Distribution of PDCoV-host interactome between vRNA- and vProtein-binding datasets.

proteins were ST-enriched (S2C Fig). To identify host proteins shared or unique among RNA viruses, we integrated ChIRP-MS datasets from SARS-CoV-2, ZIKV, DENV, and EBOV. Surprisingly, we found that nearly 40% (141/357) of vRNA-binding proteins were identified in other databases, while host proteins binding to viral RNA in these databases were derived from distinct cell lines. This finding underscores the critical role of these shared host proteins in viral infections (Fig 2E). Cluster analysis of these 141 shared host-binding proteins revealed two conserved major clusters (RBPs and translation), which also predominated in PDCoV RNA-binding protein databases. In contrast, multiple RNA viruses hijack host cells by binding to diverse metabolic enzymes for reprogramming. Additionally, RNA viruses may employ similar mechanisms to maintain the cytoskeleton, thereby facilitating viral replication (S3A Fig). Notably, network analysis identified 22 universally shared host factors (S2 Table), including a large HnRNP family (HNRNPAB, HNRNPM, SYNCRIP, HNRNPA2B1, HNRNPC, HNRNPH1, HNRNPU, HNRNPL), two DNA helicases (DDX3X, DHX9), and two SR transporters (SRSF1, SRSF3).

**Global analysis of PDCoV protein and its host protein interactome.** WB analysis of input and output samples from AP-MS experiments revealed that all vProteins were expressed in PK1 cells, with nsp5 and S proteins detectable only in output samples after bead-based enrichment due to their low abundance in input samples (Fig 2H). Pearson correlation analysis using atrMS showed inter-replicate correlations ranging from 0.41 to 0.99, confirming consistent vProtein clustering (S3B Fig and S3 Table). High-confidence PPIs identified via SAINTexpress (BFDR ≤ 0.05, spectral count ≥ 2) revealed 419 host interactors for 20 vProteins, where nsp13/S had the most interactors and nsp15 the fewest (Fig 2F). Notably, certain HIPs were shared across multiple vProtein interactomes; for instance, RPS11 and RPS25 (core ribosomal proteins involved in translation and immune regulation) appeared in 12 vProtein interaction networks (Fig 2G) [25]. Approximately half (204/419) of HIPs interacted with multiple vProteins, indicating complex vProtein-host relationships (S3 Table). Cytoscape-based visualization depicted vProteins as square nodes and HIPs as circular nodes (S3C Fig). The interaction network implicated HIPs in DNA helicase activity, membrane trafficking, transcriptional initiation factors (TIFs), protein transport, and RBPs. Interestingly, AAA+ATPases—a family mediating ATP-dependent substrate binding, folding, or degradation—were identified [26]. Gene Ontology (GO) enrichment analysis (top 5 GO-BP terms per vProtein) revealed functional divergence: nsp4/nsp8 were enriched in p53 signaling, nsp5 is primarily involved in energy metabolism and DNA replication, nsp9 regulated ubiquitin-dependent ERAD pathways, nsp7 mediated ubiquitin ligase binding, nsp12 specifically participates in SRP-dependent transmembrane translocatio, while nsp3/nsp13/S/M/NS7 participated in intracellular transport. Multiple vProteins modulated RNA-associated translation pathways to enhance viral replication (Fig 2I and S3 Table).

Immunofluorescence of affinity-tagged PDCoV ORFs in PK1 cells assessed vProtein localization and morphological impacts. Phalloidin staining evaluated cytoskeletal integrity, given coronavirus-induced cytoskeletal disruption. Cytoplasmic dominance with minor nuclear distribution was observed for nsp6, nsp8, nsp12, M, N, and S proteins. S protein expression induced pronounced morphological disruption, consistent with its role in viral entry and immune evasion (e.g., syncytium formation in IBV) [27,28]. Nuclear-predominant localization occurred for nsp2, nsp5, nsp10, nsp14, and nsp15, with nsp5 co-localizing with actin. As the main protease (Mpro), nsp5 cleaves viral polyprotein pp1ab during early infection, suppresses stress granule (SG) formation and RLR signaling to promote replication [29]. Uniform nucleocytoplasmic distribution characterized nsp7, nsp9, nsp13, nsp16, and NS6 (S4 Fig). In this section, we established a PDCoV-host interaction database comprising 671 host factors: 102 interacted with both vRNA and vProteins, 252 bound exclusively to vRNA, and 317 were vProtein-specific (Fig 2J).

## Global network mapping of PDCOV-host protein interactions

To construct a comprehensive network of PDCoV-host interactions, we performed pathway analysis on all identified interacting proteins and generated a global interaction map using Cytoscape and BioRender (Fig 3 and S4 Table). Coronavirus infection requires the viral S protein to bind host cell surface receptors, followed by viral entry via multiple pathways. Previous studies have shown that PDCoV enters cells via clathrin-mediated endocytosis [30,31], and we identified four clathrin vesicle transport-related proteins (AP1M1, DBNL, PICALM, and MYO6), with PICALM previously implicated in HIV entry [32]. After internalization, viral particles traffic from early endosomes to late endosomes, where we observed enrichment of multiple endosomal-associated proteins, some of which interacted with both vRNA and vProteins (e.g., MSN, EZR, HSPD1, PRDX3, and RDX). Upon viral RNA release, PDCoV hijacks host transcriptional and translational machinery for replication, inducing extensive host reprogramming. Notably, vRNA predominantly bound ribosomal proteins, whereas vProteins preferentially interacted with host factors involved in protein metabolism, energy metabolism, and amino acid metabolism, many of which localized to mitochondria. The significant enrichment of TCA cycle-related proteins among vProtein interactors suggests distinct organelle tropism for vRNA and vProteins post-entry. Multiple cellular pathways were implicated in viral replication, including: TIFs, transmembrane transporter activity, ubiquitination-related proteins and autophagy-associated host factors. Additionally, we identified signaling pathway-related proteins, particularly those involved in: MAPK-AKT signaling, PPAR signaling pathway, cGMP-PKG signaling pathway and Antigen processing and presentation. Interestingly, numerous hnRNP family proteins interacted with vProteins, highlighting their potential role in PDCoV infection.

For viral RNA replication, nuclear entry is required to generate gmRNA, sgmRNA, and progeny RNA. We observed enrichment of RNA-processing proteins, including those involved in: RNA splicing, RNA modification and poly(A)-binding proteins. Furthermore, p53-associated DNA damage repair proteins were significantly enriched. Viral assembly occurs in double-membrane vesicles (DMVs) derived from the endoplasmic reticulum (ER), followed by Golgi-mediated maturation and release. We found that vRNA co-opts vesicle transport proteins (e.g., COPB1, TMED10, and LAMP1)—previously linked to SARS-CoV-2-induced DMV formation. COPI proteins, which mediate retrograde transport from the Golgi to the ER [33,34], were also implicated, consistent with reports that SARS-CoV cycles through the ER-Golgi intermediate compartment (ERGIC) before budding [35]. Finally, viral infection disrupts cellular architecture [36], as evidenced by interactions between viral components and microtubule-stabilizing cytoskeletal proteins. In conclusion, PDCoV infection recruits numerous host factors, subverts multiple cellular pathways, and remodels the intracellular environment to facilitate viral replication.

## Viral-host interaction network reveals key cellular pathways hijacked during infection

Viral-host protein interactions play pivotal roles in multiple stages of viral infection. To understand how positive-sense RNA viruses evolve to interact with hosts and identify conserved infection pathways, we performed comparative analyses of interaction networks from: PDCoV vRNA/vprotein-host interactomes, SARS-CoV-2 vRNA/vprotein-host interactomes [9], ZIKV/DENV vRNA-host interactomes and Human picornavirus/rhinovirus (RV) vRNA-host interactomes. Focusing on PDCoV interactions, we observed significant enrichment of proteins involved in translation, RNA replication, and viral trafficking (S4 Table). Following cytoplasmic entry, viral RNA must engage host translation apparatus. While PDCoV and SARS-CoV-2 showed divergent ribosomal subunit preferences (potentially reflecting α- vs δ-coronavirus tropism), other positive-strand RNA viruses shared PDCoV's ribosomal protein profile. TIFs exhibited quantitative differences: PDCoV preferentially recruited EIF2B2, EIF3H and EIF5, whereas EIF3E/F, EIF4A1/A3 and EIF4B/G1/G2 were conserved across all viruses. The RNA replication requires the binding of multiple host proteins, we found that helicases and hnRNP family members were prominently enriched. Notably, HNRNPA3 and SYNCRIP interacted with both RNA and proteins of PDCoV/SARS-CoV-2, and bound ZIKV/DENV/RV RNAs - representing rare universally shared host factors. Previous studies demonstrate HNRNPA3 suppresses PEDV replication via PI3K/Akt/JNK signaling and ZNF135-mediated SREBF1 inhibition [37]. Recent studies have highlighted both the physical association of Rab GTPase family members with viral proteins

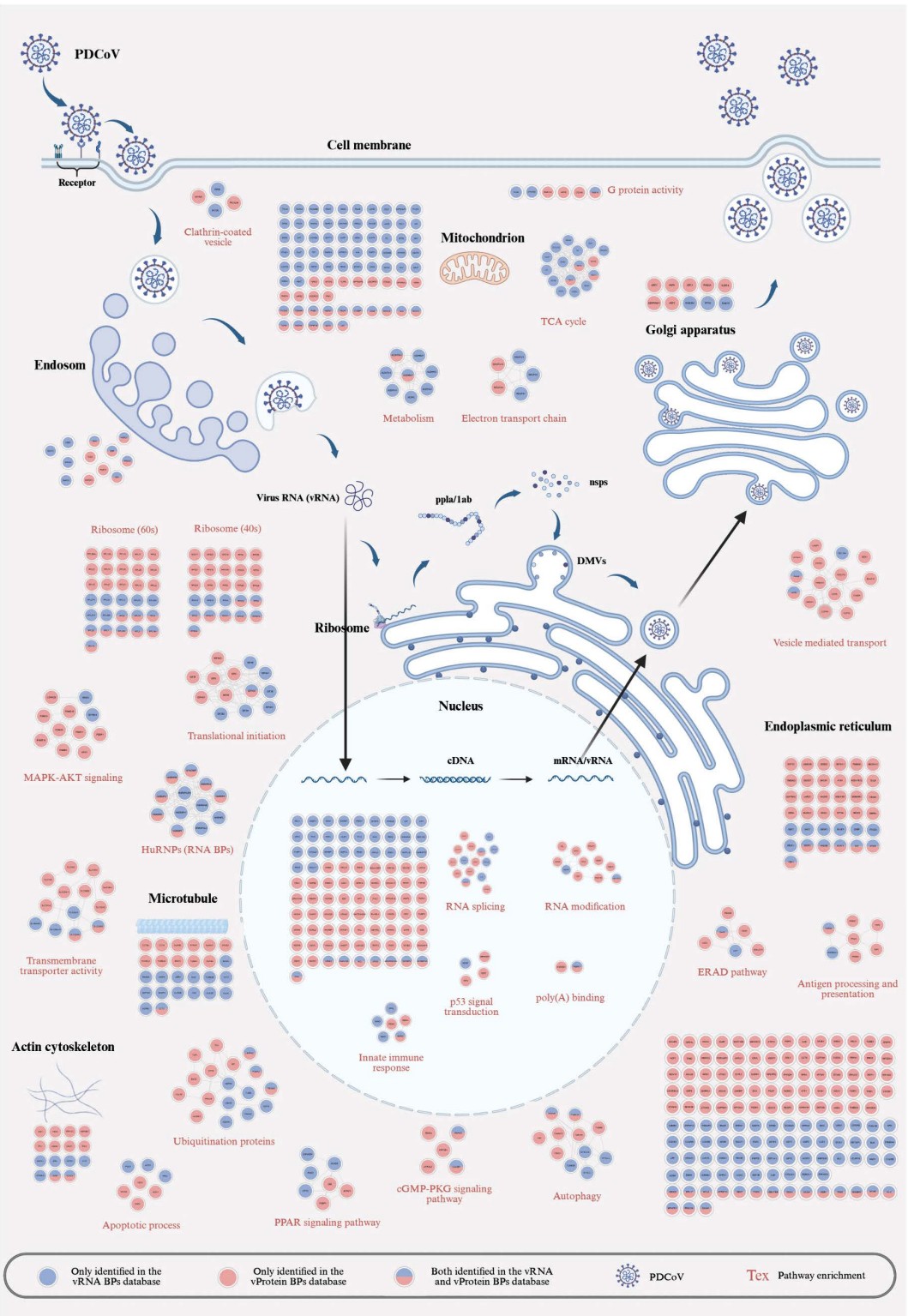

**Fig 3. Comprehensive atlas of host factors interacting with PDCoV vRNA and vProtein.** Circular nodes represent host proteins (color-coded by vRNA/vProtein binding status), with cellular compartments/pathways indicated by colored labels. The image was created in BioRender. Jun, W. (2025) https://BioRender.com/nkm0euc.

and their functional importance in the temperature-dependent life cycle of coronaviruses [13]. Our data corroborate these observations, revealing nine Rab proteins within the PDCoV interactome. Notably, RAB10 demonstrated interactions with all tested viruses, consistent with its established role in mediating intracellular vesicular trafficking of PEDV [38]. Solute carriers (SLCs) represent the largest transporter protein family in the human genome and constitute the second largest transmembrane protein family after G-protein coupled receptors (GPCRs), governing the uptake and release of diverse small molecules and metabolites [39]. CRISPR/Cas9 screening revealed SLC35A1 as an essential host factor for PDCoV infection [40], though it was absent from our interaction dataset. Furthermore, the virus-specific SLC recruitment patterns: SARS-CoV-2 preferentially engages SLC16A1, SLC3A2, and SLC9A3R17, RV primarily recruits SLC25A family members7, and ZIKV exhibits a distinct SLC interaction profile compared to other viruses.

To systematically characterize changes in PDCoV-host interacting proteins during infection, we performed 4D-DIA quantitative proteomic analysis on PDCoV-infected PK1 cells. Principal component analysis (PCA) revealed distinct clustering between infected and uninfected cells (Fig 4A). Using thresholds of P-value < 0.05/0.01 (significant/highly significant) and fold change (FC) ≥ 1.2 or ≤ 1/1.2, we identified 78 significantly up-regulated and 215 down-regulated proteins (S5 Table), indicating a global host proteome shift toward reduced expression under viral stress (Figs 4C and S5A-C). Surprisingly, only six PDCoV-interacting proteins exhibited expression changes: SAFB and COL4A1 are up-regulated; CEP192, HNRNPH1, MYL6 and EIF2B2 are down-regulated (S5D Fig). We analyzed these six proteins in both the vRNA-host interaction protein database and the vProtein-host interaction protein database. The results revealed that COL4A1, CEP192, HNRNPH1, and EIF2B2 exclusively bound to vRNA, while SAFB specifically interacted with vProtein. Notably, MYL6 was identified in both protein databases. Given the minimal overlap between interaction partners and differentially expressed proteins, we investigated whether their enriched pathways were functionally linked. GO analysis revealed that the up-regulated proteins were associated with RNA binding and protein transport, while the down-regulated proteins participated in amino acid metabolism, ubiquitination, and ER trafficking (S5E-F Fig). KEGG pathway analysis further highlighted significant alterations in DNA damage repair, autophagy, immune response, endocytosis and TNF signaling (S5G-H Fig). Notably, these perturbed pathways—including ubiquitination, transcription, signal transduction, and vesicular transport—overlapped with those enriched among PDCoV-binding proteins (S5 Table). These results suggest that while PDCoV interacts broadly with host protein families, infection directly modulates the expression of only a limited subset. Next, RT-qPCR analysis of PDCoV-infected IPEC-J2 cells confirmed that most host interactors exhibited reduced mRNA levels, whereas MAP4, MCC1, PC, MARTR3, TRIM28, and GATM were significantly up-regulated (Fig 4D). However, siRNA-mediated knockdown experiments demonstrated that depletion of FUBP1 enhanced PDCoV-N expression, while knockdown of SYNCRIP, EZR, RALY, SND1, or PSPC1 suppressed viral replication (Fig 4E–F). Network analysis revealed that hnRNP family members were shared across multiple viral interactomes, implicating them as a potential universal pathway exploited by viruses. Strikingly, siRNA knockdown of SYNCRIP, a conserved hnRNP member, significantly inhibited PDCoV replication, prompting further investigation into its antiviral mechanism.

## SYNCRIP acts as a host restriction factor with PDCoV infection

To investigate the role of SYNCRIP during PDCoV infection, we first examined its mRNA expression levels in four porcine cell lines, including two intestinal epithelial lines (IPEC-J2 and IPI-2I). RT-qPCR analysis revealed that PDCoV infection significantly up-regulated SYNCRIP mRNA in ST, PK1, and IPEC-J2 cells, whereas IPI-2I cells exhibited an opposite trend (Fig 5B). Further WB analysis demonstrated that viral infection resulted in elevated SYNCRIP protein expression in cell lines including ST, PK1, and IPEC-J2 (Figs 5B and S6B-C). We generated a Flag-tagged SYNCRIP overexpression construct (S6D Fig) and transfected PK1 cells with increasing plasmid concentrations. WB and RT-qPCR confirmed a dose-dependent increases in viral protein and mRNA levels (Fig 5C–D), a phenotype recapitulated in IPEC-J2 cells (S6E–F Fig). Then, three siRNAs targeting SYNCRIP were designed, with siRNA-SYNCRIP-2# exhibiting the highest knockdown efficiency (Fig 5E–F). Titrating siRNA-SYNCRIP-2# in PK1 cells revealed a dose-dependent decrease in viral

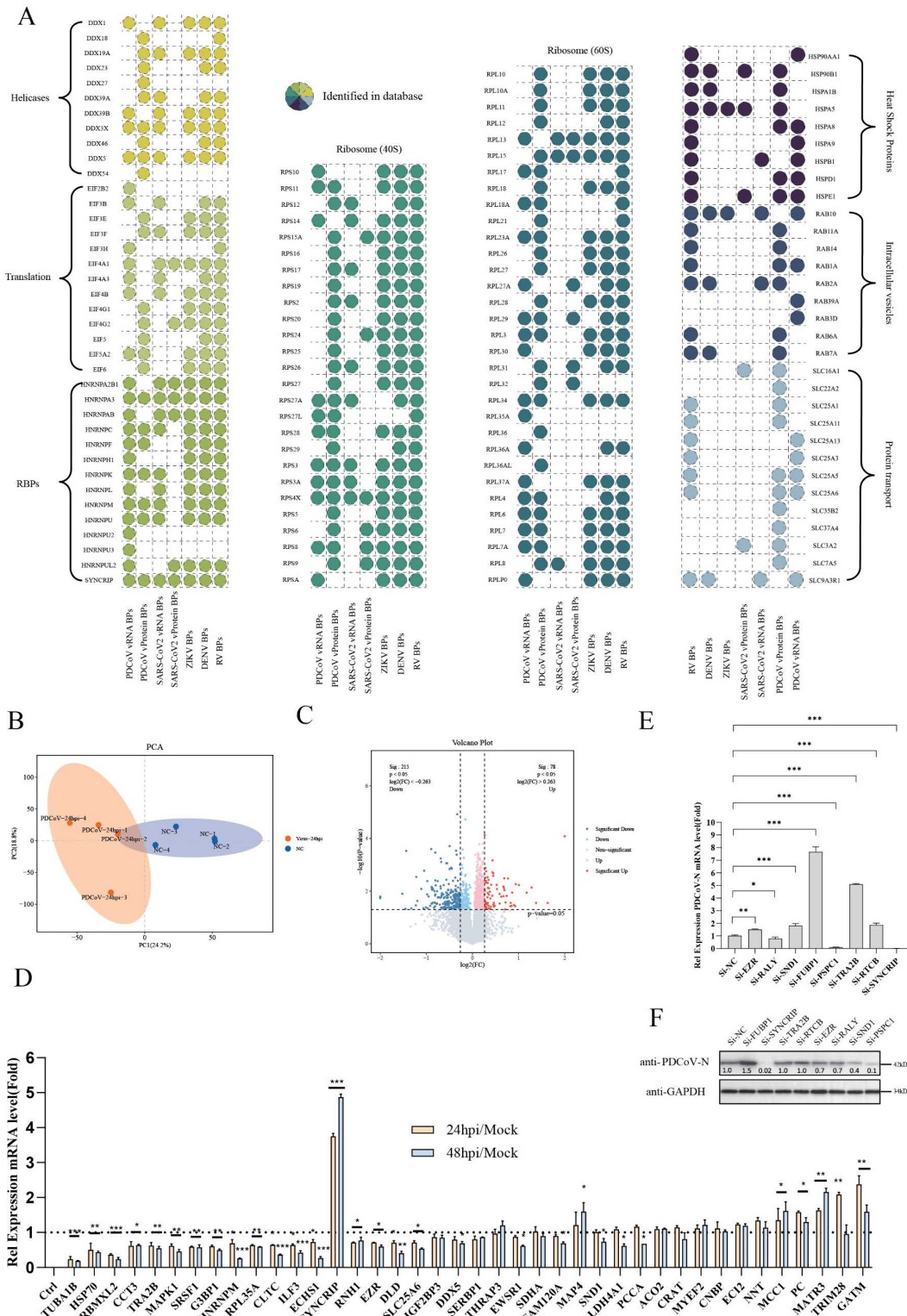

**Fig 4. (A) Left: Enrichment of DDX family, transcription factors, and hnRNP family across RNA virus-host interactomes.** Middle: Ribosomal protein (RPS/RPL) preferences among RNA viruses. Right: Viral preferences for HSPs, vesicle trafficking Rabs, and SLC transmembrane proteins. (B) PCA of proteomic profiles from LLC-PK1 cells at 24hpi with PDCoV. (C) Differentially expressed proteins (DEPs) screening in PDCoV-infected PK1 cells.

(D) RT-qPCR validation of mRNA level changes for 40 PDCoV-interacting factors post-infection. (E) Impact of siRNA knockdown (8 host factors) on viral mRNA replication (24hpi). (F) Effect of siRNA knockdown on viral N protein production (24hpi).

protein, mRNA, and supernatant viral titers (Fig 5G–I), corroborated in IPEC-J2 cells (S6G–H Fig). To further characterize SYNCRIP's role, we generated SYNCRIP-KO ST cell lines via CRISPR-Cas9 (S6I Fig and S6 Table). Eight monoclonal KO lines were isolated by limiting dilution, all maintaining >80% viability (Fig 5J–M). WB analysis showed that seven of eight KO lines (excluding sgRNA1-KO#2) significantly reduced PDCoV-N protein expression (Fig 5N). Notably, two KO lines (ST-SYNCRIP-sgRNA1-KO#4 and ST-SYNCRIP-sgRNA2-KO#1) also impaired PEDV replication (S6J Fig), suggesting a conserved antiviral mechanism across coronaviruses. To delineate the specific stage of viral replication affected by SYNCRIP, we performed time-course experiments in SYNCRIP-KO ST cell lines. RT-qPCR analysis confirmed that SYNCRIP deficiency did not impair viral attachment or internalization (S6K Fig). However, comparative analysis at 3, 6, 9, 12, and 24hpi revealed that four SYNCRIP-KO lines exhibited significantly reduced viral RNA levels compared to wild-type (WT) ST cells starting at 6hpi (Fig 5O), indicating SYNCRIP's predominant role in the replication phase. Reintroduction of Flag-tagged SYNCRIP into KO cell lines partially restored viral replication, with ST-SYNCRIP-sgRNA2#1 achieving complete rescue to WT levels by 12hpi (Fig 5P). As an RBP involved in splicing, SYNCRIP's impact on viral RNA dynamics was investigated [41]. SYNCRIP depletion reduced both gmRNA and sgmRNA levels, with RdRP activity (measured by 5'UTR and nsp2 expression) mirroring the reduction in N gene expression (S6L–N Fig). The absence of SYNCRIP led to a replication defect of the entire viral genome. Importantly, we found that the gmRNA/sgmRNA ratio decreased in SYNCRIP-KO cells (Fig 5Q), suggesting SYNCRIP modulates viral RNA processing. Finally, SYNCRIP contains three RNA recognition motifs and an IW-AP domain (S6O Fig).

To identify the critical domains of SYNCRIP required for viral replication, we generated a series of domain-deletion mutants (Fig 5R). WB analysis confirmed proper expression of all mutant constructs in vitro. Functional complementation assays in two independent SYNCRIP-KO cell lines revealed that full-length SYNCRIP partially restored viral replication and all four domain-deletion mutants failed to rescue the replication defect at early infection stage (6hpi). Similarly, full-length SYNCRIP restored approximately 70% of viral replication capacity and domain-deletion mutants showed minimal rescue activity (<20% recovery) at later time-point (12hpi). These results demonstrate that the intact structure of SYNCRIP is essential for its proviral function and multiple domains likely cooperate to facilitate efficient viral replication in ST cells.

## SYNCRIP competitively interacts with the PDCoV N protein, effectively shielding it from HUWE1-mediated ubiquitination and subsequent proteasomal degradation

To elucidate the role of SYNCRIP in viral replication, we examined its enrichment across various viral proteins. Notably, SYNCRIP exhibited significant enrichment with the N protein (S3 Table), suggesting a potential functional interaction. Subsequent immunofluorescence analysis in ST cells revealed pronounced cytoplasmic co-localization between SYNCRIP and the PDCoV N protein (Fig 6A), which was further confirmed by Co-IP assays (Fig 6B). Using our panel of SYNCRIP domain-deletion mutants, we identified that the RRM1 domain partially mediates this interaction, as its deletion attenuated (but did not abolish) N protein binding (Figs 6C and S7A). Given prior evidence that SYNCRIP overexpression enhances viral replication, we investigated its mechanism via N protein binding. In vitro co-transfection with increasing concentrations of SYNCRIP plasmid resulted in a dose-dependent elevation of N protein expression (Fig 6D). Furthermore, SYNCRIP overexpression significantly reduced the ubiquitination level of PDCoV N (S7B Fig), implying SYNCRIP may facilitate viral replication by impairing N ubiquitination. Interestingly, since SYNCRIP is not a deubiquitinating enzyme, we employed IP-MS to identify its interactors (Fig 6E). The interactome analysis revealed no deubiquitinases but identified the E3 ubiquitin ligase HUWE1, which was also found in the PDCoV N interactome (S7C Fig). HUWE1 (also known as UBE1, HECTH9, ARF-BP1, MULE), a large HECT-family E3 ligase comprising 4374 amino acids [42], was detected

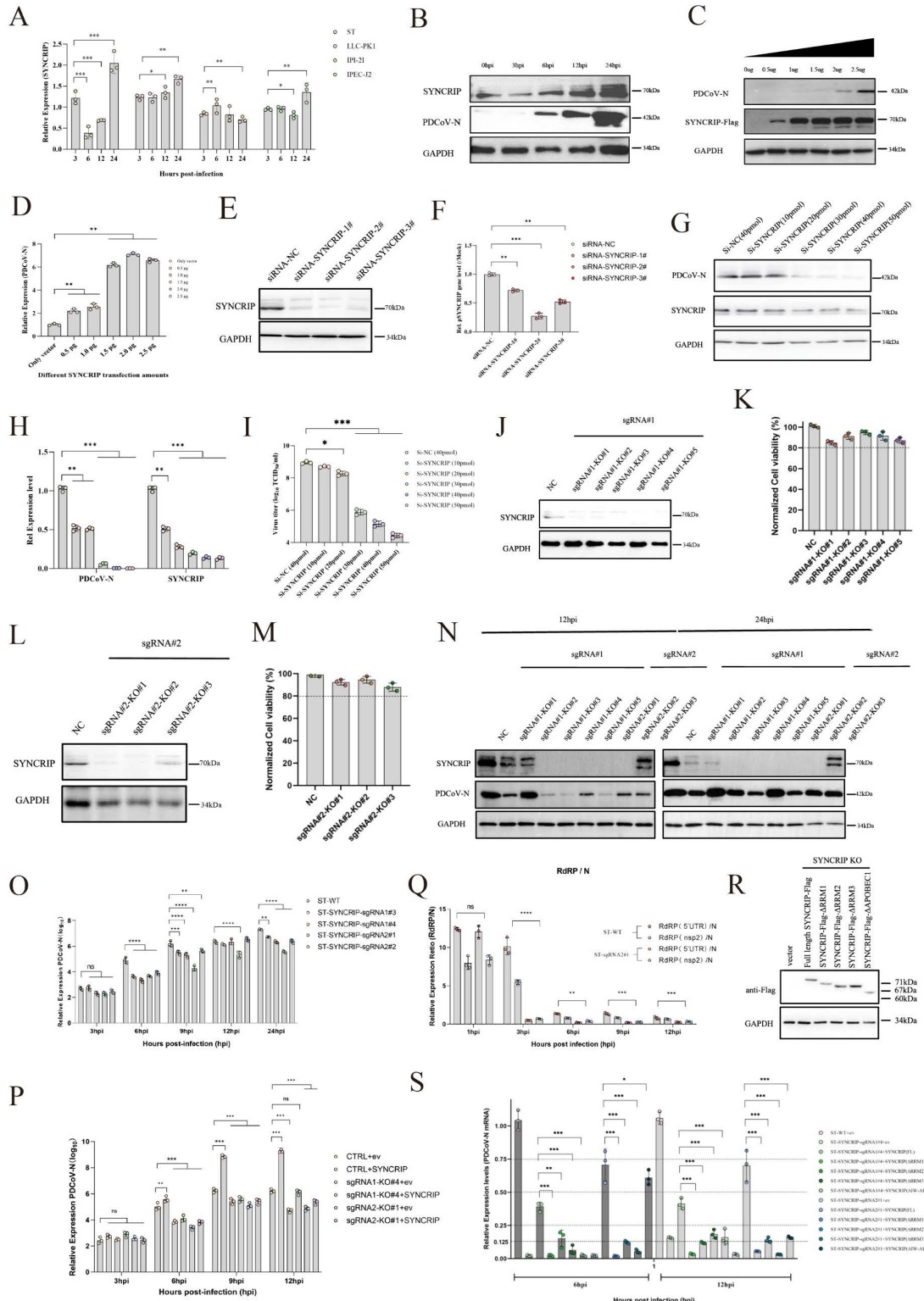

**Fig 5. (A) mRNA expression changes of SYNCRIP in four porcine cell lines following PDCoV infection.** (B) Protein expression dynamics of SYNCRIP in ST cells post PDCoV infection. (C) Effect of SYNCRIP overexpression at varying concentrations on viral N protein expression in ST cells. (D) Impact of SYNCRIP overexpression on viral mRNA levels in ST cells. (E) Protein knockdown efficiency assessment of three SYNCRIP-targeting

siRNAs. (F) mRNA knockdown efficiency of three SYNCRIP-targeting siRNAs. (G) Viral protein expression in ST cells treated with gradient concentrations of siRNA-SYNCRIP-2#. (H) Viral mRNA levels in ST cells following siRNA-SYNCRIP-2# treatment at different doses. (I) Viral titers in supernatants of siRNA-SYNCRIP-2#-treated ST cells. (J) SYNCRIP protein levels in five sgRNA#1-mediated stable knockout cell lines. (K) Cell viability assessment of five sgRNA#1 SYNCRIP knockout clones. (L) SYNCRIP expression in three sgRNA#2-generated stable knockout cell lines. (M) Viability of three sgRNA#2 SYNCRIP knockout cell lines. (N) PDCoV protein expression at 12/24 hpi across eight SYNCRIP knockout clones. (O) Temporal viral mRNA expression profiles in SYNCRIP knockout cells. (P) Viral mRNA replication rescue by SYNCRIP (Flag-tagged) complementation in knockout cells. (Q) gRNA/sgRNA ratio alterations in SYNCRIP knockout cells. (R) Schematic of SYNCRIP domain-deletion mutants. (S) Viral mRNA replication upon complementation with SYNCRIP domain mutants in knockout cells.

in SYNCRIP IP eluates (Fig 6F). Confocal microscopy confirmed the SYNCRIP-HUWE1 interaction (S7E Fig), and IP using PDCoV N likewise demonstrated its binding to HUWE1 (Fig 6G). As HUWE1 primarily recognizes substrates via its HECT domain, we co-transfected increasing amounts of HECT domain plasmid with PDCoV N. This resulted in an inverse correlation between N protein expression and HUWE1-HECT levels (Fig 6H). Overexpression of HUWE1-HECT markedly attenuated viral replication (S7F Fig up), while HUWE1 knockdown using 100nmol/mL B18622 in PK1 cells enhanced viral replication (S7F Fig down). To determine the degradation pathway utilized by HUWE1-HECT for PDCoV N, we employed inhibitors targeting the two primary intracellular protein degradation pathways: the ubiquitin-proteasome system (UPS) and the autophagy-lysosome pathway. Inhibition of the UPS with MG-132 abolished HUWE1-HECT-mediated degradation of PDCoV N, whereas inhibitors of autophagy-lysosomal degradation (CQ and 3-MA) had no rescuing effect (Fig 6I). Moreover, HUWE1-HECT overexpression significantly enhanced PDCoV N ubiquitination (Fig 6J). Site-directed mutagenesis was performed on the four cysteine residues within the HUWE1 HECT domain, known to rely on Cys304 for ubiquitin transfer (Fig 6K). Co-expression of these mutants with PDCoV N revealed that the C304A mutation specifically restored N protein expression, while other point mutants (C147A, C330A) did not (Fig 6L). Assessment of N ubiquitination levels showed that mutations at C147A, C304A, and C330A all abrogated HUWE1-HECT's ability to ubiquitinate N (Fig 6M), demonstrating that C147, C304, and C330 are critical for HUWE1-HECT's ubiquitin ligase activity, but C304 is specifically required for recognition and ubiquitination of the PDCoV N protein.

As a core protein of the hnRNP family, and its presence was identified in multiple RNA-host protein interaction datasets, suggesting it may serve as a critical host factor in cross-viral infections. Subsequent conservation analysis of SYNCRIP across humans, pigs, chickens, cattle, mice, bats, and monkeys revealed >99% similarity among most species, except for chickens, which exhibited multiple variations (S6 Table); this indicates SYNCRIP likely performs conserved functions during viral infections. Consequently, given SYNCRIP's regulatory role on coronavirus N protein, we constructed N proteins from three additional alpha-coronaviruses (PEDV, TGEV, and SADS-CoV[Swine Acute Diarrhea Syndrome Coronavirus]) and one gammacoronavirus (Infectious Bronchitis Virus, IBV)in vitro. WB demonstrated higher expression levels for PEDV and IBV N proteins, but reduced expression for TGEV and SADS-CoV N proteins (Fig 6N), while Co-IP confirmed interactions between SYNCRIP and the N proteins of all four coronaviruses (Fig 6O). We then validated the regulatory effect of HUWE1-HECT on N proteins from other coronaviruses: HUWE1-HECT overexpression reduced their expression (S7G Fig), and this inhibition was rescued by treatment with the ubiquitin-proteasome inhibitor MG132 (S7H-K Fig). Similarly, HUWE1-HECT overexpression significantly enhanced the ubiquitination levels of viral N proteins (S7L Fig). As HUWE1 interacts with both SYNCRIP and viral N proteins, we investigated SYNCRIP's specific role in HUWE1-mediated infection through in vitro co-transfection of SYNCRIP, HUWE1, and PDCoV N protein. Co-IP assays revealed diminished binding of HUWE1 to PDCoV N protein upon SYNCRIP overexpression (Fig 6P). Additionally, we observed that SYNCRIP overexpression significantly diminished the ubiquitination capacity of HUWE1-HECT domain toward PDCoV N protein (S7M Fig). Using previously established monoclonal cell lines with stable SYNCRIP knockout (ST-SYNCRIP-sgRNA1#4 and ST-SYNCRIP-sgRNA2#1), we observed that SYNCRIP knockout further augmented the inhibitory capacity of HUWE1-HECT against the virus (Fig 6Q). Collectively, these findings demonstrate that SYNCRIP likely functions as a key protein in coronavirus infection.

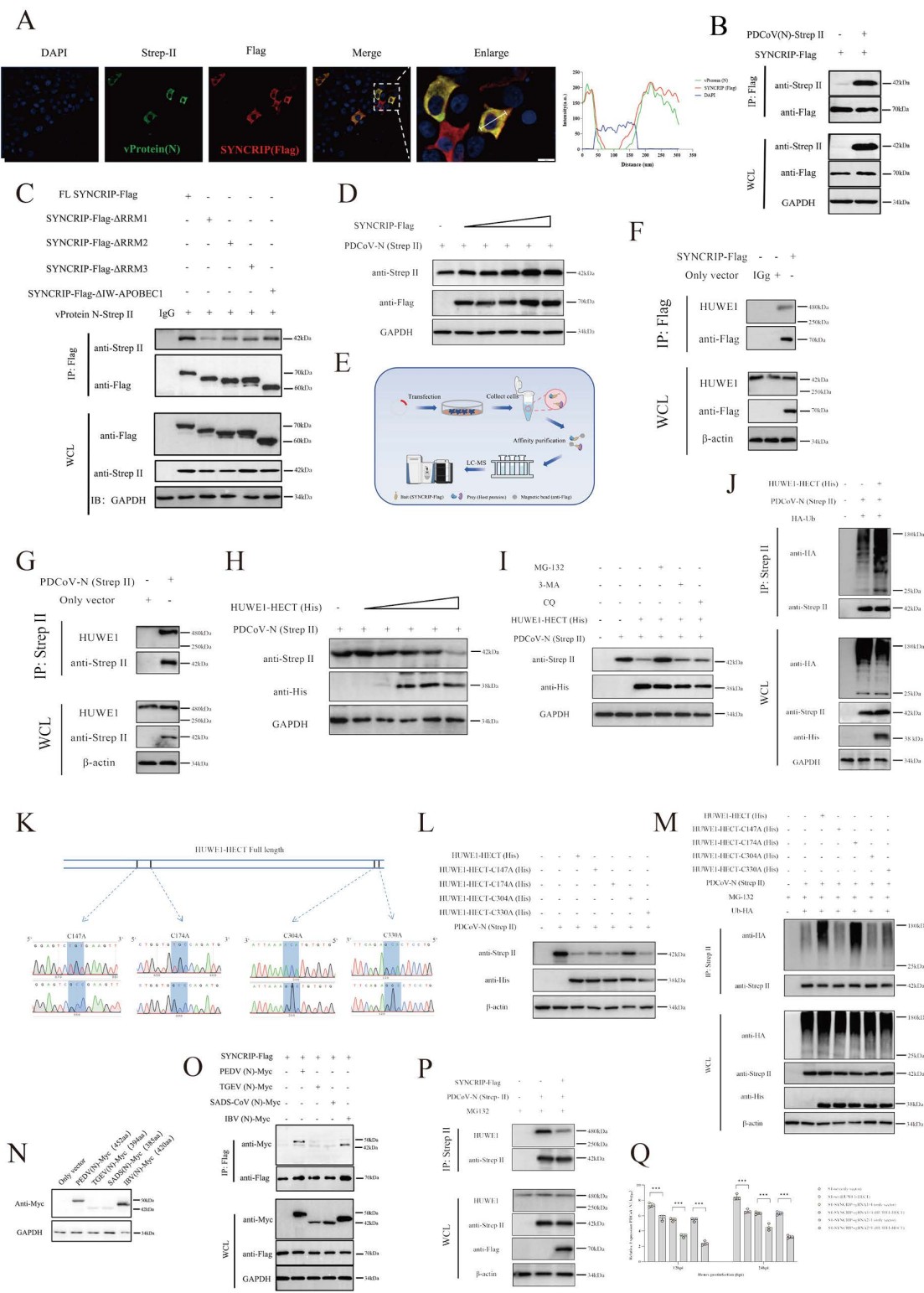

**Fig 6. (A) Confocal microscopy showing cytoplasmic colocalization of SYNCRIP with PDCoV N protein.** (B) Co-IP validation of SYNCRIP-N protein interaction. (C) Domain mapping of SYNCRIP-N interaction using deletion mutants. (D) Dose-dependent effect of SYNCRIP (Flag-tagged) overexpression on N protein levels. (E) Workflow for IP-MS identification of SYNCRIP-interacting host proteins. (F) Co-IP analysis of the interaction

between SYNCRIP and endogenous HUWE1. (G) Co-IP detection of PDCoV N binding to endogenous HUWE1. (H) Dose-dependent effect of HUWE1-HECT (His-tagged) overexpression on N protein levels. (I) WB analysis of N protein expression under proteasome (MG132) and lysosome (CQ/3-MA) inhibition. (J) Ubiquitination status of PDCoV N protein upon HUWE1-HECT overexpression. (K) Schematics and sequencing results of HECT domain catalytic mutants. (L) Effect of HUWE1-HECT catalytic mutant (E3 ligase-dead) on N protein levels. (M) Ubiquitination analysis of N protein with HUWE1-HECT catalytic mutant. (N) Expression constructs for N proteins of PEDV/TGEV/SADS-CoV/IBV. (O) Co-IP verification of SYNCRIP interactions with heterologous coronavirus N proteins. (P) Co-IP assessment of PDCoV N-HUWE1 association following SYNCRIP overexpression. (Q) Enhanced viral suppression by HUWE1-HECT upon SYNCRIP knockout. Illustration from NIAID NIH BioArt Source (bioart.niaid.nih.gov/bioart/).

## Targeting SYNCRIP with small-molecule inhibitors effectively suppresses PDCoV/PEDV/IBV replication

Given the crucial regulatory role of SYNCRIP in viral infection, we explored its potential as an antiviral target by screening small-molecule inhibitors against its RRM2 domain (S8A Fig). Molecular docking analysis identified five top-scoring compounds: Neamine, Isoforsythiaside, Deapi-platycodin D3, Ginsenoside Ra2, and Kukoamine B. Cytotoxicity ($CC_{50}$) and inhibitory concentration ($IC_{50}$) assays in PK1, ST, and IPEC-J2 cell lines revealed that Isoforsythiaside and Kukoamine B exhibited superior antiviral activity and cost-effectiveness (Figs 7A, 7B and S8B-G), prompting further characterization. Immunofluorescence assays demonstrated dose-dependent inhibition of PDCoV proliferation by both compounds in ST cells (Fig 7C), with consistent results observed in PK1 and IPEC-J2 cells (S8H Fig). WB and RT-qPCR analyses confirmed significant reduction of viral N protein expression (Fig 7D) and mRNA levels (Fig 7E) following treatment. Mechanistic studies using time-of-addition assays (Fig 7F) revealed that neither compound directly inactivated virions (S8I Fig), but both moderately reduced viral adsorption in PK1 and IPEC-J2 cells (S8J Fig) without affecting internalization (S8K Fig). Crucially, both inhibitors profoundly impaired viral replication (Fig 7G), aligning with SYNCRIP's established role in this stage. Furthermore, the broad-spectrum activity was further validated against PEDV, showing consistent suppression of viral mRNA and protein levels across all cell lines (Fig 7H-I). Evaluation in avian HD11 cells demonstrated high $CC_{50}$ values for both compounds (Fig 7J), with Isoforsythiaside showing superior potency in reducing IBV mRNA and protein expression (Fig 7K-L).

## Isoforsythiaside demonstrates significant antiviral efficacy in vivo

To evaluate the in-vivo antiviral potential of Isoforsythiaside, we conducted infection experiments using 3-day-old piglets. Fifteen piglets were divided into three groups (n = 5/group), with Group 3 receiving oral administration of 5 mg/kg Isoforsythiaside for 3 consecutive days prior to PDCoV challenge (Fig 8A). Necropsy revealed that PDCoV infection induced severe intestinal pathology characterized by thinning/transparent intestinal walls, emphysematous distension, and yellow fluid accumulation - all of which were markedly attenuated in Isoforsythiaside-treated animals (Fig 8B). Notably, the treatment showed no adverse effects on normal growth. While PDCoV infection caused up to 15% weight loss (peaking at 7dpi) due to appetite suppression from intestinal damage, Isoforsythiaside significantly mitigated this effect (Fig 8C). Serological analysis demonstrated earlier seroconversion in treated groups, with PDCoV-specific IgG detectable in 2/3 samples by 3dpi versus 7dpi in controls (Fig 8D). Viral shedding was delayed until 14dpi in treated piglets compared to consistent detection from 3dpi in controls (S9A Fig). Histopathological examination of PDCoV-infected intestinal tissues by HE staining revealed severe tissue damage in all infected samples, manifested as villous atrophy, blunting, and even detachment. Further pathological observations demonstrated extensive inflammatory cell infiltration and hemorrhage in the jejunum and duodenum of PDCoV-infected piglets, while the ileum and rectum primarily exhibited goblet cell reduction. In contrast, intestinal tissues from Isoforsythiaside-treated piglets remained relatively intact, with preserved villous architecture despite persistent inflammatory cell infiltration in the jejunum. No significant pathological changes were observed in control group piglets (Fig 8E).

Given PDCoV's broad tissue tropism, we quantified viral loads in various tissues at different timepoints. The heart and liver were not major infection sites. Substantial viral RNA was detected in the lungs, kidneys, and all four intestinal

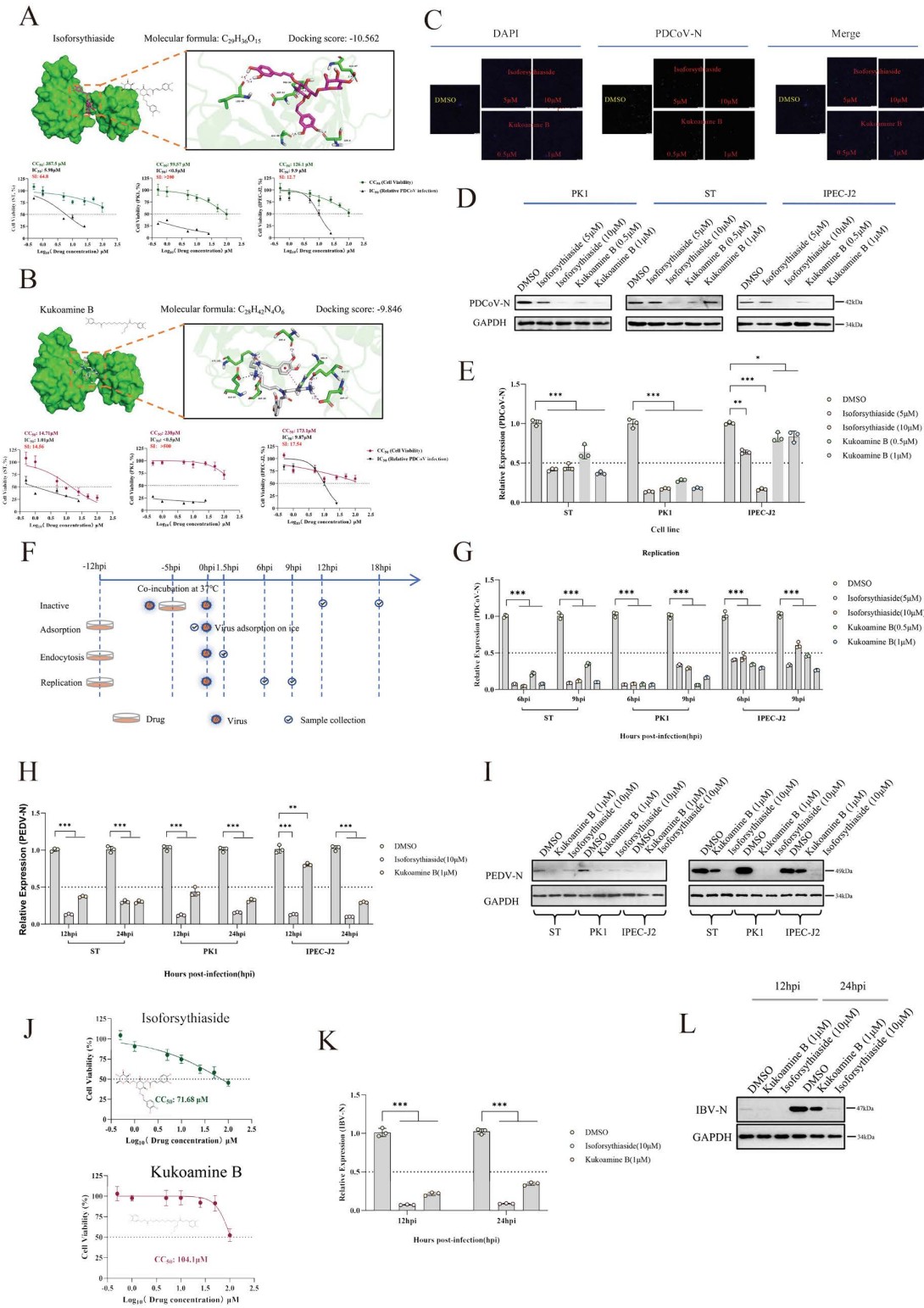

**Fig 7. (A) Top: Molecular docking model of Isoforsythiaside with SYNCRIP RRM2 domain.** Bottom: Cytotoxicity (CC$_{50}$) and inhibitory concentration (IC$_{50}$) of Isoforsythiaside in PK1, ST, and IPEC-J2 cells. (B) Top: Molecular docking simulation of Kukoamine B with SYNCRIP RRM2. Bottom: CC$_{50}$ and IC$_{50}$ determination for Kukoamine B in three cell lines. (C) IFA quantification of PDCoV N protein expression in ST cells treated with gradient

concentrations of Isoforsythiaside/Kukoamine B. (D) Western blot analysis of viral N protein in three cell lines following compound treatment. (E) PDCoV N mRNA levels in drug-treated cells across concentrations. (F) Schematic of experimental design for assessing antiviral stage specificity. (G) Replication-phase inhibition by compounds in three cell lines (dose-response). (H) Suppression of PEDV N mRNA replication by both compounds. (I) Dose-dependent inhibition of PEDV N protein expression. (J) Cytotoxicity profiles ($CC_{50}$) in HD11 cells. (K) IBV N mRNA replication inhibition by treatments. (L) Compound-mediated suppression of IBV N protein expression.

segments, with viral loads in lungs, kidneys and rectum increasing over time. Viral RNA levels in ileum, jejunum and duodenum peaked at 7dpi before declining. Isoforsythiaside treatment significantly reduced viral loads across all infection stages in jejunum, rectum and duodenum, while only decreasing loads at 7dpi and 14dpi in the other three tissues (lungs, kidneys and ileum) (Fig 8F). The production of inflammatory cytokines in target tissues represents part of the innate immune response to viral infection [43]. qRT-PCR analysis of cytokine levels (IL-6, IL-8, TNF-α and IFN-α) in jejunum, ileum, rectum and duodenum showed that PDCoV infection significantly elevated cytokine mRNA levels in all intestinal tissues. Isoforsythiaside treatment reduced cytokine levels across all intestinal tissues, although it did not completely suppress the cytokine elevation, demonstrating partial mitigation of virus-induced inflammatory cytokine upregulation (S9B Fig). These promising results support host-directed antiviral drug discovery as an important strategy for PDCoV prevention and control.

## Discussion

As an emerging coronavirus with cross-species infection potential, PDCoV poses significant challenges to the livestock industry due to its broad transmission capacity [44]. PDCoV infection involves complex virus-host interactions, making the understanding of these dynamics crucial for elucidating regulatory mechanisms of viral replication and pathogenicity. In this study, we conducted a large-scale proteomic investigation and established a comprehensive PDCoV-host protein interaction network comprising 671 host proteins through integration across multiple timepoints and cell lines. Among these, 354 proteins interacted with vRNA and 419 with vProtein, participating in diverse biological processes including metabolic pathways, translation, transcription, vesicle trafficking, and ubiquitination regulation. Subsequent integration of the virus-host interaction network identified several host factors essential for viral entry, replication, or dissemination. Notably, multiple host proteins were found to interact with both vRNA and vProtein, with 102 proteins classified into this dual-interaction group (Fig 9). Furthermore, our PDCoV-host interactome mapping revealed SYNCRIP as a novel host restriction factor against PDCoV infection.

   RNA viruses consist primarily of infectious genomic RNA and functional encoded proteins. While previous studies have focused either on vRNA-centric or vProtein-centric virus-host interaction mapping [12,45,46], such approaches are incomplete because viral infection requires coordinated cooperation between vRNA and vProtein—neither alone is sufficient to produce infectious virions [8,47]. Therefore, interaction datasets derived solely from vRNA or vProtein fail to fully capture how viruses hijack and reprogram host machinery. Here, we employed ChIRP-MS (vRNA-centric) and AP-MS (vProtein-centric) to construct a comprehensive PDCoV-host interaction network. Many identified host proteins, including GLUD1, PDHX, PHB2, GOT1, GOT2, LDHA, PDHB, G3BP1, ISG15, ANXA2, ANXA5, HDAC6, and STAG2, have been previously implicated in PDCoV replication [48–53], yet a substantial number remain uncharacterized. Genome-wide CRISPR-Cas9 screening identified multiple host restriction factors for PDCoV replication, such as SLC35A1, TMEM41B, CTSL, CTSB, HSP90AB1, and C16orf62 [40,54–57], which were absent from our interaction network. Most of these host factors are membrane proteins or membrane-localized transporters, while viral infection requires the binding of the S protein to cell surface receptors to initiate the replication program [58]. Although CRISPR-Cas9 can effectively identify host factors involved in early viral infection, it fails to monitor key host factors responsible for reprogramming host life processes during viral infection. These factors may restrict viral infection not through direct binding to vRNA/vProtein but by disrupting or reinforcing virus-host protein connections, subsequently amplifying their effects through cascading regulatory

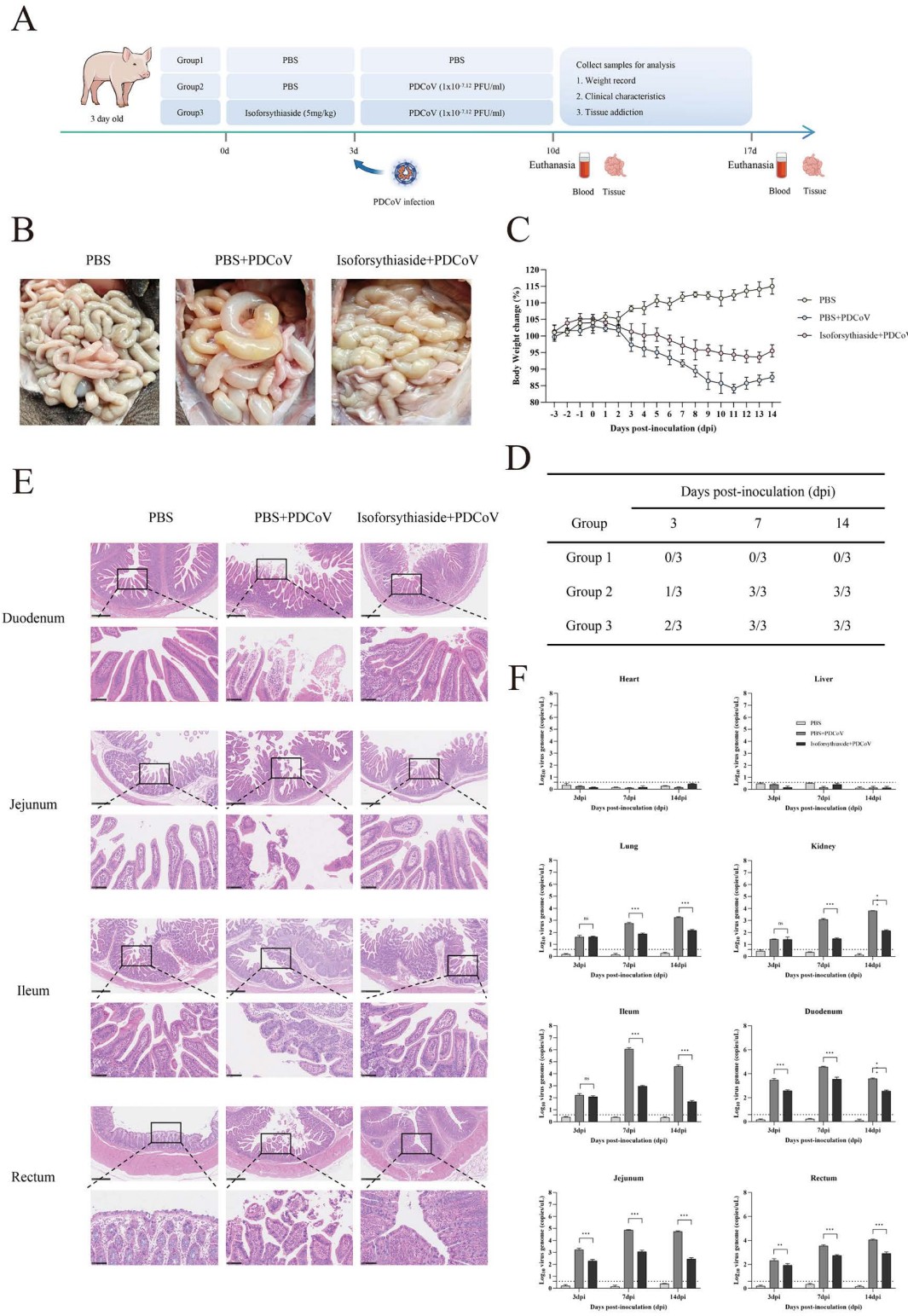

**Fig 8. (A) Experimental workflow for evaluating Isoforsythiaside efficacy in PDCoV-infected piglets (3-day-old).** (B) Histopathological changes in intestinal tissues of treated piglets. (C) Body weight dynamics of experimental groups. (D) Serum IgE antibody levels post-treatment. (E) HE-stained pathological sections of duodenum, jejunum, ileum, and rectum. (F) PDCoV replication levels in multiple organs (heart, liver, lung, kidney, intestinal segments). Illustration from NIAID NIH BioArt Source (bioart.niaid.nih.gov/bioart/).

PLOS Pathogens

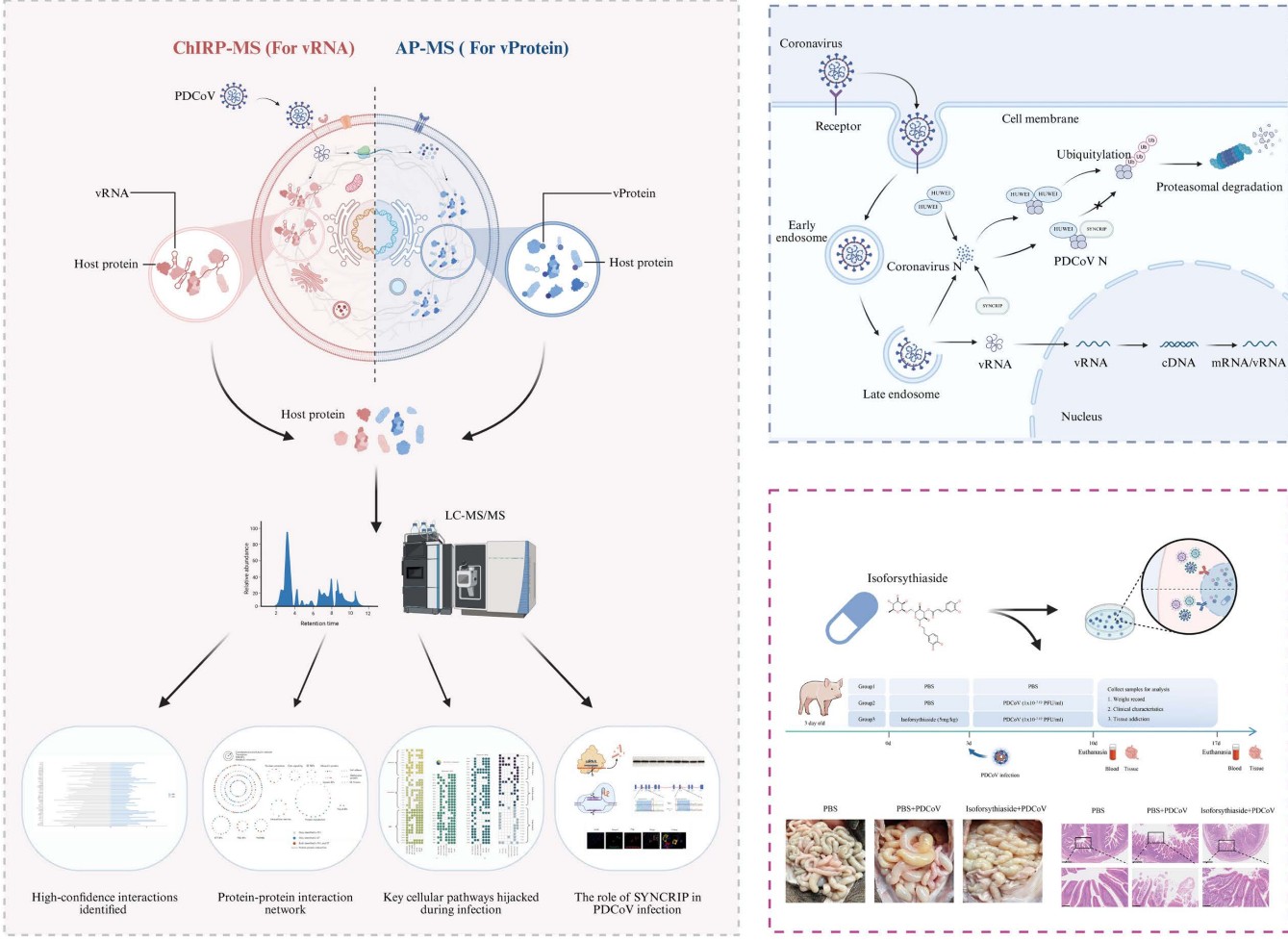

**Fig 9. Schematic of the PDCoV-host protein interaction network mapped by ChIRP-MS and AP-MS.** Host proteins interacting with PDCoV were identified by centering on vRNA and vProtein, respectively. The interaction network revealed that the host protein SYNCRIP promotes viral infection by inhibiting HUWE1-mediated ubiquitination of the coronavirus N protein. Both in vivo and in vitro studies demonstrate that SYNCRIP is a potential antiviral target. The image was created in BioRender. Jun, W. (2025) https://BioRender.com/mq46stf.

networks. This highlights the necessity of complementary methodologies in studying virus-host interactions. Viruses rely on host cellular machinery for replication, and from an evolutionary perspective, they typically avoid inducing widespread changes in host protein expression, as such perturbations could prematurely trigger cell death—an unfavorable outcome for viral propagation. Instead, viruses subtly co-opt multiple host proteins, leveraging their functional networks. Any virus-induced perturbations in host homeostasis are distributed across the PPI network [59]. Our proteomic analysis revealed that fewer than 1% (6/671) of host proteins exhibited direct expression changes during viral replication, yet the functional roles of these DEPs aligned closely with those of PDCoV-interacting proteins, demonstrating the virus's sophisticated ability to manipulate host systems.

Distinct viruses may utilize similar intracellular pathways for replication. By integrating virus-host interaction datasets from multiple viruses, we identified common strategies employed by viruses to hijack host machinery for transport and replication. The SARS-CoV-2 pandemic has heightened researchers' interest in virus-host interaction patterns. Multiple viruses may utilize similar infection pathways, and comparative studies of different viral interaction

proteomes have accelerated the development of broad-spectrum antiviral drugs. RNA-host interaction proteomes generated by other RNA viruses were included in the comparison. Collectively, RNA viruses appear to reprogram host cells by hijacking common pathways. For instance, both PDCoV and SARS-CoV-2 interact with numerous metabolic enzymes and RBPs, including RBP SND1 which has been demonstrated to bind negative-sense SARS-CoV-2 RNA and coordinate nascent viral RNA synthesis through NSP9 recruitment [46]. A hallmark of RNA virus infection is the formation of DMVs on the endoplasmic reticulum [60,61]. Our analysis revealed multiple vesicle trafficking-related proteins (LAMP1, TMED10, COPA, COPB1) as common viral targets. However, due to divergent host adaptation strategies among viruses, they exhibit distinct protein preferences during host cell infection [62]. The proteins encoded by eIFs genes play central roles in the initiation stage of eukaryotic protein synthesis. While multiple viruses bind to similar eIF4 complexes to initiate translation, different viruses additionally hijack other eIFs genes to either promote viral protein expression or inhibit normal host cellular metabolism. Our investigations revealed that PDCoV demonstrates strong preference for eIF2B, eIF3H, and eIF5, whereas SARS-CoV-2 preferentially interacts with eIF3F and eIF4A3. Similarly, PDCoV primarily relies on RPL18, RPL21, RPL32, RPL35, and RPL36 to regulate ribosomal binding, while other viruses exhibit higher affinity for RPL11 and RPL24. These findings further underscore the differential strategies employed by distinct viruses during host infection. Notably, several hnRNP family proteins not only persistently interact with both vRNA and vProtein during PDCoV infection but also engage with other viruses. As canonical RBPs recognizing specific sequences and maintaining nucleic acid homeostasis, hnRNPs have been shown to regulate viral RNA replication and facilitate immune evasion [63–66]. We focused on the HnRNP family protein SYNCRIP (hnRNP Q) and demonstrated that viral replication efficiency decreased following SYNCRIP knockdown using either siRNA or CRISPR-Cas9. Conversely, SYNCRIP overexpression enhanced viral replication. These findings establish SYNCRIP as an essential host factor for coronavirus proliferation. Although SYNCRIP's RNA recognition via RRM domains is well-characterized (e.g., its antiviral role through PPV NS1 mRNA interaction) [14], its effect upon overexpression prompted investigation into vProtein interactions. This study reveals that SYNCRIP interacts with the N proteins of multiple coronaviruses and reduces their ubiquitination levels, thereby enabling them to evade host ubiquitin-mediated degradation systems. As SYNCRIP lacks intrinsic ubiquitination/deubiquitination domains, it likely functions by recruiting deubiquitinating enzymes or disrupting the ubiquitination machinery targeting the N protein. Multiple E3 ubiquitin ligases recognize coronavirus N proteins as substrates, while TRIM28 and CUL5 mediate N-protein ubiquitination in PEDV and SARS-CoV-2 respectively [67,68]. We discovered that SYNCRIP does not recruit any deubiquitinating enzymes (DUBs), but rather directly binds the E3 ubiquitin ligase HUWE1. HUWE1 reportedly induces K48-linked polyubiquitination of ORF9b (SARS-CoV-2) [69], ORF3 (MERS-CoV) [70], and HIV Gag-Pol [71], attenuating interferon suppression [72]. Similarly, HUWE1 interacts with the PDCoV N protein. We found that SYNCRIP overexpression disrupts this interaction, significantly impairing HUWE1-PDCoV N binding. Subsequent SYNCRIP depletion further enhanced HUWE1's ability to suppress viral replication. Given SYNCRIP's cross-viral functionality, we computationally identified Isoforsythiaside as a SYNCRIP-targeting inhibitor. It effectively suppressed replication of PDCoV (δ-coronavirus), PEDV (α-coronavirus), and IBV (γ-coronavirus) in vitro, while mitigating PDCoV-induced tissue tropism and host damage in vivo. Importantly, SYNCRIP exhibits >99% amino acid conservation between humans and swine, underscoring its potential as a pan-coronaviral therapeutic target, particularly for human coronavirus outbreaks.

In summary, we have established an integrative host interactome profiling approach centered on both PDCoV RNA and encoded proteins to investigate pathogenic coronavirus infections, through which we identified conserved targeting mechanisms likely shared across other members of this viral family. Our findings highlight 671 hub proteins as potential therapeutic targets, delineate the critical role of SYNCRIP in coronavirus infection, and evaluate its potential as a pan-antiviral target. These results may inform future studies to elucidate viral pathogenesis and develop therapeutic strategies against coronaviruses.

## Methods and materials

### Ethics statement

All animal care was conducted under strict supervision by the Animal Management of Sichuan University. The experiment involving infecting the chicken with the virus was approved by the Animal Ethics Committee (AEC) of the Sichuan University (License: SCU240301001). Briefly, all the pigs were sacrificed for procuring the samples using carbon dioxide, followed by exsanguination. At the end of the experiments, the pigs were intravenously injected with pentobarbital sodium.

### Cell lines and virus

The LLC-PK1, ST, IPEC-J2, IPI-2I, and 293T cell lines were maintained in Dulbecco's Modified Eagle Medium (DMEM; Gibco) supplemented with 10% heat-inactivated fetal bovine serum (FBS; Gibco) and 1% penicillin/streptomycin (Gibco). All cell lines were confirmed to be mycoplasma-free by PCR testing prior to experiments and authenticated through microscopic morphological examination. All procedures involving infectious viruses were performed in a Biosafety Level 2 (BSL-2) facility approved by the Institutional Biosafety Committee of Sichuan University. The PDCoV strain SCCZ18 (GenBank accession no. MT985156) was isolated and preserved in our laboratory. Viral titers in cell culture supernatants were determined by 50% tissue culture infectious dose ($TCID_{50}$) assay and calculated using the Reed-Muench method.

### Comprehensive identification of vRNA-binding proteins via ChIRP-MS

LLC-PK1 and ST cells were seeded at a density of $1 \times 10^{4.5}$ cells per well in a 6-well plate. At 12 hours post-inoculation (hpi), the cells were infected with PDCoV at MOIs of 0.01, 0.1, and 1. Cell morphology was observed at 24 hpi and 48 hpi, and total cellular proteins and RNA were collected for PDCoV proliferation assessment. PDCoV-targeted probes were designed online using the Stellaris platform (https://www.biosearchtech.com/stellaris). By default, each probe was designed to bind approximately 200–400 bp of viral RNA, evenly covering the entire transcript. The complete probe sequences are provided in S1 Table. The oligonucleotides were synthesized by Synbio Technologies, China, and modified with 3′ biotin-TEG.

The ChIRP-MS procedure was primarily conducted as described previously [73]. LLC-PK1 or ST cells were seeded at $1 \times 10^6$ cells per T150 flask. At 12 hpi, the cells were infected with PDCoV at an MOI of 0.01 or mock-infected. Three T150 flasks were used for each condition. After removing the supernatants, the cells were washed twice with PBS, fixed with formaldehyde for 30 minutes, and quenched with a final concentration of 0.125 M glycine. The cells were then washed three times with PBS and stored at -80°C. The cells were thawed on ice and lysed in ten volumes of lysis buffer (50 mM Tris-HCl pH 7.0, 10 mM EDTA, 1% SDS). The cell lysates were sonicated and centrifuged at 12,000 rpm/min, and the supernatants were retained. The supernatants were pretreated with 50 µL of washed MyOne C1 beads (Thermo Scientific, 65001), and the beads were subsequently removed. Next, two volumes of CHIRP hybridization buffer (750 mM NaCl, 1% SDS, 50 mM Tris-HCl pH 7.0, 1 mM EDTA, 15% formamide) and 10 µL of a 100 µM ChIRP probe pool (an equimolar mix of all probes) were added. The samples were hybridized at 37°C for 12 hours with rotation. Subsequently, 300 µL of washed MyOne C1 beads were added, and the samples were incubated at 37°C for 45 minutes with rotation. The enriched material was collected on the beads using a magnetic stand, and the beads were washed five times with 1 mL of ChIRP wash buffer (2x NaCl-citrate [SSC, ThermoFisher Scientific], 0.5% SDS). After washing, 1% of the sample was used for RNA extraction. To elute the enriched proteins, the beads were collected on a magnetic stand, resuspended in ChIRP biotin elution buffer (12.5 mM biotin, 7.5 mM HEPES pH 7.9, 75 mM NaCl, 1.5 mM EDTA, 0.15% SDS, 0.075% sarcosine, and 0.02% sodium deoxycholate), and mixed at 25°C for 20 minutes and at 65°C for 15 minutes with shaking. The elution was transferred to a new tube, and the beads were eluted again. The two elution fractions were combined. The protein elution was concentrated using the trichloroacetic acid-acetone precipitation method and dissolved in 1x LDS buffer with 20 mM DTT. The samples were boiled at 95°C for 30 minutes with occasional stirring for reverse crosslinking.

The protein samples were subjected to SDS-PAGE electrophoresis and stained with a silver staining kit (Sangon Biological, China) for visualization. The protein bands were excised from the gel, destained, desalted, pH-adjusted, and digested overnight with trypsin. The digested peptides were extracted with 50% acetonitrile and 0.1% formic acid and analyzed by MS (Thermo Scientific Q Exactive).

To determine the vRNA recovery rate of ChIRP-MS, 1% of the sonicated cell lysate was used as the input sample. During the final wash, 10 μL of wash buffer containing 1% of the beads was taken as the elution fraction for RNA quantification. The input sample and elution fraction were suspended in 100 μL of PK buffer (10 mM Tris-HCl pH 7.0, 100 mM NaCl, 0.5% SDS, 1 mM EDTA). The samples were incubated at 50°C for 45 minutes and then at 95°C for 10 minutes with mixing. Finally, RNA was extracted using 300 μL of TRIzol LS reagent according to the manufacturer's instructions. Reverse transcription was performed using the PrimeScript RT kit (TAKARA, RR047A) to synthesize cDNA. To distinguish viral RNA from host RNA, we primarily employ specific primer sets targeting distinct genomic regions. Viral RNA detection relies on primers designed against the ORF1ab gene, while host RNA serves as an internal control amplified by GAPDH-targeted primers.

### Identification of binding proteins of PDCoV-encoded proteins by AP-MS

The IDT codon optimization tool (https://www.idtdna.com/codonopt) was utilized to perform codon optimization on all Nsps, S, E, M, N, NS6, and NS7 proteins in the PDCoV genome, with the removal of internal EcoRI and BamHI sites. Start and stop codons were added to Nsps 2–16 as needed, Kozak sequences were inserted before each start codon, and 2×-Strep tags with linkers were appended to either the N-terminus or C-terminus. Subsequently, $1\times10^7$ PK1 cells were seeded into each 15 cm petri dish and transfected with 15 μg of individual Strep-tagged expression constructs 12 hours later. The total plasmid amount was standardized to 15 μg using an empty vector and complexed with PEI 3000 transfection reagent (SignaGen Laboratories) at a plasmid-to-reagent ratio of 1:3 μg:μl, as recommended by the manufacturer. Cells were collected after more than 24 hours and washed three times with PBS. After each step, cells were centrifuged at 1000 rpm/min for 5 minutes at 4°C. The cell pellets were frozen on dry ice and stored at −80°C. For each bait, n = 3 independent biological replicate samples were prepared for affinity purification. Frozen cell pellets were thawed on ice for 15–20 minutes and suspended in 1 ml of IP lysis buffer with freshly added 1% PMSF (Thermo Scientific, 36978). The lysates were centrifuged at 13,000 rpm/min for 15 minutes at 4°C to pellet debris, and 50 μl of the lysate was retained as the input sample. Subsequently, 60 μl of pre-cleaned MagStrep "type3" beads (IBA Lifesciences, 2–4090) were added, and the mixture was incubated overnight at 4°C with rotation. The beads were then washed three times with 1 ml of 1×Buffer W wash buffer (IBA Lifesciences, 2–1003). After the final wash, the supernatant was removed, and 80 μl of 1×Buffer BXT (IBA Lifesciences, 2–1042) was added. The beads were adsorbed using a magnetic rack, and the supernatant was collected. Six times protein buffer was added to the supernatant, which was then incubated at 95°C for 10 minutes with thorough mixing. Finally, MS identification was performed according to the ChIRP-MS protocol.

### Identification of high-confidence interactions (HCI) with vRNA/vProtein

All raw mass spectrometry results were processed for protein identification and alignment using MaxQuant. For ChIRP-MS samples, the R package "Differential Enrichment Analysis for Proteomics Data" (DEP; https://rdrr.io/bioc/DEP/man/DEP.html) was employed to define the "high-confidence" interactome for each PDCoV infection condition (24hpi/48hpi, and LLC-PK1/ST). DEP has its own data preprocessing procedures, thus for this analysis, the DEP default workflow was used to directly perform filtering, normalization, and estimation on the MaxQuant outputs. Significantly different/extremely significantly different enriched protein sets were defined using cut-off values of $\log_2$ fold change (FC)> 0 and adjusted p-value ≤ 0.05/0.01, respectively, by comparing infected cells after PDCoV RNA pull-down with mock-infected (control) cells subjected to the same treatment. Additionally, host interactome protein libraries generated by ChIRP-MS for other RNA viruses, including SARS-CoV-2[2], ZIKV [74], DENV [74], and EBOV [12], were collected to search for conserved vRNA-binding factors across viral cell entries.

For AP-MS samples, the tool (http://proteomics.fi/) in conjunction with the SINTeractome Significance Analysis Tool (SAINT) express version 3.6.0 was used as the statistical method to identify specific high-confidence interactions from AP-MS data [75]. Twenty-four GFP control runs (12 N/C-terminal MAC-GFP, 6 C-terminal MAC-GFP with a myristoylation signal sequence, and 6 C-terminal MAC-GFP with a nuclear localization signal sequence) were utilized as control counts for each match. HCI were defined as those with an estimated protein-level Bayesian False Discovery Rate (BFDR) ≤ 0.01. Furthermore, the CRAPome database was employed with a cut-off frequency of ≥ 20% (≥ 82), except for those with a mean spectral count FC ≥ 3, to eliminate potential false positives [76]. Files processed by SAINT contained quantitative information on bait-prey interactions and were uploaded to the tool (https://prohits-viz.org/) for correlation analysis and generation of detailed bait-bait comparisons [77].

### Visualization of host interaction factors of vRNA/vProtein using cytoscape

All identified host factors binding to vRNA/vProtein were uploaded to STRING (https://cn.string-db.org) to construct PPIs. The PPI data were then imported into Cytoscape 3.8 for visualization of the PPI network. For gene classification analysis [78], we utilized the DAVID bioinformatics resource annotation (https://davidbioinformatics.nih.gov) [79] to obtain Gene Ontology (GO) terms and Kyoto Encyclopedia of Genes and Genomes(KEGG) pathway analysis for all interacting factors. The significance p-values for GO terms and KEGG pathways were calculated using the Fisher's exact test and corrected using FDR (FDR < 0.05).

### Proteomic analysis of PDCoV-infected PK1 cells

Protein identification was performed using 4D data-independent acquisition (4D-DIA) on PK1 cells before and after PDCoV infection. Cells infected at MOI = 0.01 were harvested at 24 hpi alongside uninfected controls. Total proteins were extracted, subjected to trypsin digestion and peptide desalting, followed by LC-MS analysis. Raw DIA data were processed using Spectronaut Pulsar 18.4 (Biognosys) software for database searching. Proteins meeting the following criteria were retained: (1)unique peptides ≥ 1; (2) Valid values present in ≥ 2 samples; (3) At least one group with valid value proportion ≥ 50%. Missing values in groups with ≥ 50% valid value proportion were imputed using the group mean. Remaining missing values were replaced with half the sample minimum value. Median normalization and $\log_2$ transformation were applied to obtain credible proteins. Differential protein expression (DEPs) was evaluated using two parameters: (1) FC: $\log_2(FC)$ = (mean of experimental group) - (mean of control group); (2) P-value derived from Student's t-test. DEPs were defined as those satisfying, FC ≥ 1.5 or ≤ 1/1.5 (0.667) and P-value < 0.05. Subsequent analyses of DEPs included PCA, hierarchical clustering, GO enrichment, and KEGG pathway analysis.

### Microscopy analysis of vProtein subcellular localization/co-localization

ST/PK1 cells were seeded on coverslips in 12-well plates. When cells reached approximately 50% confluency, they were transfected with specific expression plasmids. At 24 hours post-transfection, cells were fixed with 4% paraformaldehyde for 20 minutes and permeabilized with 0.1% Triton X-100 at room temperature for 15 minutes. Subsequently, cells were blocked with 5% bovine serum albumin (BSA) for 1 hour, followed by incubation with primary antibodies for 2 hours. After three washes with PBS containing 0.1% Tween-20 (PBST), cells were stained with Alexa Fluor-conjugated secondary antibodies for 1 hour. Nuclei were counterstained with 4′,6-diamidino-2-phenylindole (DAPI) at room temperature for 10 minutes. Imaging was performed using an Olympus SpinFV-COMB confocal microscope (Olympus, Japan).

### Cell transfection with siRNA and plasmids

siRNAs targeting specific genes were designed and synthesized by Synbio Technologies (China). The constructed plasmid vectors—including pLVX-SYNCRIP(Flag-tag)-Puro, pLVX-SYNCRIPΔRRM1(Flag-tag)-Puro, pLVX-SYNCRIPΔRRM2(Flag-tag)-Puro, pLVX-SYNCRIPΔRRM3(Flag-tag)-Puro, pLVX-SYNCRIPΔIW-AP(Flag-tag)-Puro,

pLVX-PDCoV-N(StrepII-tag)-Puro, pLVX-HUWE1-HECT(His-tag)-Puro, pLVX-HUWE1-HECT-C147A(His-tag)-Puro, pLVX-HUWE1-HECT-C174A(His-tag)-Puro, pLVX-HUWE1-HECT-C304A(His-tag)-Puro, pLVX-HUWE1-HECT-C330A(His-tag)-Puro, pLVX-PEDV-N(Myc-tag)-Puro, pLVX-TGEV-N(Myc-tag)-Puro, pLVX-SADS-CoV-N(Myc-tag)-Puro, pLVX-IBV-N(Myc-tag)-Puro, and all eukaryotic mammalian expression plasmids for vProteins—were transfected into target cells. Plasmid DNA was complexed with PEI 3000 transfection reagent (SignaGen Laboratories) at a 1:3 μg:μl plasmid-to-reagent ratio according to the manufacturer's recommendations. After 15-minute incubation at room temperature, complexes were added dropwise to cells.

Cells were harvested 24 hours post-transfection, washed three times with PBS, and lysed in an appropriate volume of RIPA lysis buffer (Beyotime, China) for 30 minutes on ice. Lysates were centrifuged at 12,000rpm/min for 20 minutes at 4°C. Supernatants were collected and stored at -80°C.

### Co-immunoprecipitation (Co-IP) and western blotting (WB) analysis

To validate protein interactions via Co-IP, co-transfected cells were washed with ice-cold PBS and lysed in 300 μL of ice-cold RIPA buffer supplemented with protease inhibitor cocktail (P8340, Sigma) for 30 minutes. After centrifugation at 12,000 rpm/min for 20 minutes, supernatants were rotated overnight at 4°C with specific antibodies. Subsequently, Protein A/G agarose beads (MCE, HY-K0230) were added and incubated for 4 hours at 4°C. Beads were collected by centrifugation at 2,500 rpm/min for 5 minutes at 4°C, washed four times with cold PBST, and eluted with SDS loading buffer after boiling for 10 minutes to release bound proteins. Standard immunoblotting procedures were then employed to analyze the proteins. For WB analysis, the eluted proteins were separated by 12.5% SDS-PAGE electrophoresis at 120V for 50 minutes and transferred to a polyvinylidene fluoride (PVDF) membrane (Millipore, USA) at 220mA for 100 minutes. The membrane was blocked with 5% skimmed milk at room temperature for 3 hours, followed by incubation with primary and horseradish peroxidase (HRP)-conjugated secondary antibodies. Detection was performed using an enhanced chemiluminescence (ECL) system (Bio-Rad, USA).

### Construction of stable knock-out(KO) cell lines using CRISPR-Cas9

Small guide RNAs were designed using the CRISPR design tool (http://crispr.mit.edu). To generate SYNCRIP- KO cells, sgRNA sequences targeting exon 2 and exon 8 of SYNCRIP were cloned into the lentiCRISPR v2 vector. According to the manufacturer's instructions, lentiCRISPR v2 vectors containing sgRNA, pSPAX2, and pMD2.G were mixed in a 1:1:2 ratio and transfected into 293T cells using Lipofectamine 2000 (Invitrogen, 11668027) to produce lentiviruses, which were then used to infect ST cells. ST cells were selected with puromycin for three days; surviving cells were trypsinized, diluted to one cell per 200 μL of DMEM medium (10% FBS), and plated into 96-well plates for clonal selection. Sanger sequencing was used to confirm cells containing nonsense mutations. The protein expression level of the KO cell line was further confirmed by WB using an anti-SYNCRIP antibody (14024–1-AP, Proteintech). To re-express SYNCRIP in SYNCRIP-KO ST cells, the SYNCRIP coding sequence was cloned into the pLVX-Puro vector and packaged into lentiviruses by co-transfecting HEK293T cells. SYNCRIP-KO cells were transfected with the packaged lentiviruses to obtain "SYNCRIP re-expression cells". Transfected cells were selected with puromycin for 4 days, and SYNCRIP re-expression was verified by Western blotting using an anti-SYNCRIP antibody.

### Screening of small molecule inhibitors targeting SYNCRIP

SYNCRIP contains three RRM domains in total. Due to the lack of a complete crystal structure of SYNCRIP, the publicly available database only includes structures for RRM2 and RRM3. RRM2 is a crucial domain regulating mRNA splicing, editing, transport, turnover, and translational control, thus it was selected as the binding pocket for small molecule drug design. The three-dimensional structure of Human SYNCRIP (PDB ID: 6KOR) was downloaded from the PDB website. The Protein Preparation Wizard module was used to add hydrogen atoms to the protein and remove water molecules.

Subsequent energy optimization was performed using the OPLS2005 force field with an RMSD cut-off of 0.30 Å. The processed protein was then used to generate a grid file using the Receptor Grid Generation module, centering on the key amino acids PHE11 and PHE55 at the binding site of the RRM2 domain of Human SYNCRIP with RNA. The box size was set to 20 Å × 20 Å × 20 Å. The Bioactive Compound Library Plus (HY-L001P, containing 24K compounds) and the Antiviral Library (HY-L0073V, containing 3.2K compounds) were used as the small molecule libraries for virtual screening. All small molecule drugs were processed through the LigPrep Module of Schrödinger software for hydrogenation, energy optimization, etc., before proceeding to virtual screening. The Virtual Screening Workflow module was employed for virtual screening, utilizing the Glide module for molecular docking. The binding ability of the protein with small molecule compounds was assessed based on the docking score (the higher the absolute value, the stronger the binding affinity between the compound and the protein). The compounds were ranked from low to high docking scores (S6 Table). The top 5 small molecule inhibitors were selected for further molecular docking and antiviral studies.

### Animal experiment protocol

Fifteen 3-day-old healthy piglets were randomly divided into three groups ($n = 5$). Each experimental group was housed in a separate room with strictly controlled environmental conditions (temperature, humidity, etc.) and standardized diet. No apparent clinical symptoms were observed in these piglets, and virus-specific PCR confirmed they tested negative for PEDV, Transmissible Gastroenteritis Virus(TGEV), PDCoV, Porcine Circovirus Type 2(PCV-2), and Porcine Reproductive and Respiratory Syndrome Virus (PRRSV). As shown in Fig 8A, Group 1 and Group 2 received oral PBS administration for three consecutive days, while Group 3 was prophylactically administered 5 mg/kg Isoforsythiaside for three consecutive days. Subsequently, Group 2 and Group 3 were orally challenged with PDCoV at $1.0 \times 10^{7.12}$ TCID$_{50}$/head, while Group 1 received an equivalent volume of PBS. Body weight and clinical symptoms were monitored daily. One piglet per group was euthanized for necropsy at 3 days post-inoculation (dpi) and 7 dpi. All remaining piglets were euthanized at 14 dpi.

### Porcine intestinal morphology (H&E staining)

Duodenum, jejunum, ileum, and rectum were collected immediately after euthanasia. Tissues were fixed in 4% paraformaldehyde, embedded in paraffin, dehydrated through an ethanol gradient series, and sectioned. For each sample, at least two 3-μm-thick sections were prepared and stained with hematoxylin and eosin (H&E). Stained sections were imaged and analyzed using a Wisleap WS-10 microscope (Zhiyue, China).

### ELISA detection of PDCoV N antibody levels

Serum antibodies were detected using an anti-PDCoV N antibody kit (Xuanzekang, China) according to the manufacturer's protocol.

### Reverse transcription quantitative real-time PCR (RT-qPCR)

For cell samples: Transfected or virus-infected/uninfected cells were collected and lysed in 300 μL TRIzol (Takara, China). Total RNA was extracted using chloroform-ethanol precipitation. RNA concentration was quantified using a Nanotrap 2000 (USA), and 1 μg total RNA was used for cDNA synthesis. RT-qPCR was performed using FastStart Universal SYBR Green Mix (Accurate Biology, Hunan, China) on a Bio-Rad real-time PCR system. qPCR was then performed with the primer sequences listed in S6 Table. Relative gene expression levels were calculated using the $2^{-\Delta\Delta Ct}$ method, with β-actin as the internal control. For tissue samples: Tissues were ground in liquid nitrogen and homogenized in 500 μL TRIzol, followed by standard RNA extraction. Viral RNA load in tissues was absolutely quantified using the standard curve equation: $y = -3.583x + 36.54$ ($R^2 = 1.0$, amplification efficiency = 90.1%).

## Statistical analysis

All data were analyzed using GraphPad Prism 5.0. Differences were evaluated for statistical significance using one-way analysis of variance (ANOVA) or Student's t test. $p < 0.05$ (*) indicates statistically significant, $p < 0.01$ (**) suggests statistically very significant, while $p < 0.001$ (***) implies statistically extremely significant.

## Supporting information

**S1 Fig. (A) MOI and time-course optimization for PDCoV SCCZ18 infection in LLC-PK1/ST cells.** (B) vRNA recovery efficiency by ChIRP-MS protocol. (C) Percentage distribution of genomic (ORF1a/b) vs. subgenomic RNA reads pre-/post-pulldown. (D) Inter-sample correlation coefficients (n = 3) at 24/48hpi in ST/PK1 cells. (E)-(H) Host binding proteins identified in LLC-PK1/ST cells at 24/48hpi. (I) Persistent host interactors across timepoints in PK1 cells. (TIF)

**S2 Fig. (A) Left: Cytoscape network of PDCoV-host interactions in PK1 cells.** Right: Pathway preferences at different infection stages. (B) Left: Host interaction network in ST cells. Right: Temporal pathway enrichment. (C) Cell line-specific differences in PDCoV-host protein interactions. (TIF)

**S3 Fig. (A) Cytoscape visualization of shared host interactors among RNA viruses (central node: vRNA; pie-chart nodes: multi-viral hits).** (B) Inter-replicate correlation coefficients (n = 3) for AP-MS-identified vProtein interactomes. (C) Cytoscape network of vProtein-host interactions (square nodes: viral proteins; color gradient by Indegree value; gray clusters: cellular pathways). (TIF)

**S4 Fig. Subcellular localization profiles of all vProteins in PK1 cells.** (TIF)

**S5 Fig. (A) SDS-PAGE analysis of protein differences between 4D-DIA proteomics samples.** (B) Number of protein hits in 4D-DIA proteomics samples. (C) Identification of differentially expressed host proteins (P-value < 0.05, FC ≥ 1.2 or FC ≤ 1/1.2) in PDCoV-infected PK1 cells. (D) Left: Heatmap clustering of DEPs expression levels (red: high; blue: low). Right: Expression changes of six PDCoV-interacting host proteins pre-/post-infection. (E) Top GO terms for DEPs (-$\log_{10}$ p-value vs. term; bar labels indicate protein counts). (F) Comparative GO enrichment (top 10 up/down-regulated terms by p-value; x-axis: ListHits/TotalHits ratio). (G) KEGG pathway analysis of DEPs (bubble size: protein count; color gradient: p-value significance). (H) Comparative KEGG enrichment (top 10 up/down-regulated pathways by p-value). (TIF)

**S6 Fig. (A) Replication kinetics of PDCoV in four porcine cell lines.** (B) Temporal SYNCRIP protein expression in PDCoV-infected LLC-PK1 cells. (C) SYNCRIP expression dynamics in infected IPEC-J2 cells. (D) IFA validation of SYNCRIP (Flag-tagged) expression. (E) Dose-dependent effect of SYNCRIP overexpression on viral N protein in IPEC-J2 cells. (F) Impact of SYNCRIP overexpression on viral mRNA replication. (G) Viral protein suppression by siRNA-SYNCRIP-2# treatment (gradient concentrations). (H) Dose-responsive inhibition of viral mRNA by siRNA-SYNCRIP-2#. (I) Sanger sequencing confirmation of exon2/exon-targeting sgRNA knock-in. (J) SYNCRIP knockout significantly inhibits PEDV mRNA replication. (K) Effect of SYNCRIP knockout on viral attachment/entry phases. (L)-(N) Impact on PDCoV N/RdRP(5'UTR)/RdRP(nsp2) mRNA replication in knockout cells. (O) Domain architecture of SYNCRIP and deletion mutant schematics. (TIF)

**S7 Fig. (A) Domain mapping of SYNCRIP-N interaction using deletion mutants (IP:Strep II).** (B) Co-IP analysis of PDCoV N ubiquitination status following SYNCRIP overexpression. (C) IP-MS identification of host ubiquitination machinery components interacting with SYNCRIP. (D) IFA verification of HUWE1-HECT (His-tagged) expression in ST cells. (E) Confocal colocalization of SYNCRIP (Flag-tagged) with endogenous HUWE1. (F) Top: Impact of HUWE1-HECT overexpression on viral replication. Bottom: Effect of HUWE1 inhibitor BI8622 on PDCoV N mRNA replication. (G) Effect of HUWE1-HECT overexpression on PEDV/TGEV/SADS-CoV/IBV N protein levels. (H)-(K) WB analysis of N protein stability under proteasome/lysosome inhibition. (L) Ubiquitination status of other coronavirus N proteins with HUWE1-HECT overexpression. (M) Co-IP assay was performed to examine the ubiquitination level of PDCoV N protein mediated by HUWE1-HECT domain upon SYNCRIP overexpression.
(TIF)

**S8 Fig. (A) Structural models of SYNCRIP RRM2/RRM3 domains.** (B)-(D) Molecular docking models of Neamine/Deapi-platycodin D3/Ginsenoside Ra2 with SYNCRIP. (E)-(G) CC50/IC50 determination for three compounds in PK1/ST/IPEC-J2 cells. (H) IFA quantification of PDCoV N suppression by compounds in PK1/IPEC-J2 cells. (I)-(K) Stage-specific inhibition of viral inactivation/attachment/entry by treatments.
(TIF)

**S9 Fig. (A) Fecal viral shedding in Isoforsythiaside-treated piglets.** (B) Cytokine (IL-6/IL-8/TNF-α/IFN-α) expression changes in intestinal segments post-treatment.
(TIF)

**S1 Table. The sequence of 109 biotinylated probes with 3' Biotin-TEG modification used for ChIRP-MS.**
(XLSX)

**S2 Table. Screening results of vRNA-host protein interactions by ChIRP-MS.**
(XLSX)

**S3 Table. Screening results of vRNA-host protein interactions by AP-MS.**
(XLSX)

**S4 Table. GO and KEGG enrichment analysis of PDCoV-host interacting proteins.**
(XLSX)

**S5 Table. Quality analysis of 4D-DIA proteomics for PDCoV infection in PK1 cells.**
(XLSX)

**S6 Table. Protein sequence differences of SYNCRIP in different species; Sequences of oligonucleotides (primers, siRNAs, and sgRNAs) used in this study; Top 20 small molecule inhibitors targeting SYNCRIP.**
(TIF)

## Acknowledgments

We thank professor Cao Yang for the support of bioinformatics analysis and this work was supported by the Animal Disease Prevention and Green Development Key Laboratory of Sichuan Province.

## Author contributions

**Conceptualization:** Rongbin Qiu, Changwei Lei.

**Data curation:** Rongbin Qiu, Changwei Lei.

**Formal analysis:** Qingcheng Yang, Yue Sun.

**Funding acquisition:** Xin Yang.

**Investigation:** Kailu Wang.

**Methodology:** Kailu Wang, Song Liu, Chengyao Hou, Yue Sun, Yiming Tian.

**Resources:** Wenjun Yan, Kailu Wang, Qinyuan Chu.

**Software:** Wenjun Yan, Yizhi Tang.

**Supervision:** Qinyuan Chu.

**Visualization:** Hao Li, Siyu Huang, Yiming Tian.

**Writing – original draft:** Wenjun Yan.

**Writing – review & editing:** Hongning Wang, Xin Yang.

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
