## [Decision Letter · Decision Letter 0]

27 Aug 2025

A comprehensive PDCoV-host proteome interaction map reveals potential antiviral targets

PLOS Pathogens

Dear Dr. Yang,

Thank you for submitting your manuscript to PLOS Pathogens. After careful consideration, we feel that it has merit but does not fully meet PLOS Pathogens's publication criteria as it currently stands. Therefore, we invite you to submit a revised version of the manuscript that addresses the points raised during the review process.

Please submit your revised manuscript within 60 days Oct 26 2025 11:59PM. If you will need more time than this to complete your revisions, please reply to this message or contact the journal office at plospathogens@plos.org. Please include the following items when submitting your revised manuscript:

We look forward to receiving your revised manuscript.

Kind regards,

Luis Martínez-Sobrido

Academic Editor

PLOS Pathogens

Alexander Gorbalenya

Section Editor

PLOS Pathogens

Editor-in-Chief

PLOS Pathogens

orcid.org/0000-0003-2946-9497

Editor-in-Chief

PLOS Pathogens

orcid.org/0000-0002-7699-2064

**Journal Requirements:**

At this stage, the following Authors/Authors require contributions: Wenjun Yan, Kailu Wang, Song Liu, Rongbin Qiu, Qingcheng Yang, Hao Li, Siyu Huang, Chengyao Hou, Qinyuan Chu, Yue Sun, Yizhi Tang, Cangwei Lei, Yiming Tian, Hongning Wang, and Xin Yang. Please ensure that the full contributions of each author are acknowledged in the "Add/Edit/Remove Authors" section of our submission form.

https://journals.plos.org/plospathogens/s/submission-guidelines#loc-parts-of-a-submission

- ® on page: 45 and 46.

Potential Copyright Issues:

- Please confirm (a) that you are the photographer of Figure 8B, or (b) provide written permission from the photographer to publish the photo(s) under our CC BY 4.0 license.

- Figures 1, 6, and 8. Please confirm whether you drew the images / clip-art within the figure panels by hand. If you did not draw the images, please provide (a) a link to the source of the images or icons and their license / terms of use; or (b) written permission from the copyright holder to publish the images or icons under our CC BY 4.0 license. Alternatively, you may replace the images with open source alternatives. See these open source resources you may use to replace images / clip-art:

5) Please ensure that the funders and grant numbers match between the Financial Disclosure field and the Funding Information tab in your submission form. Note that the funders must be provided in the same order in both places as well.

**Reviewers' Comments:**

Reviewer's Responses to Questions

**Part I - Summary**

Reviewer #1: This manuscript presents a systematic and comprehensive investigation of host-pathogen interactions between PDCoV and its host cells. The authors employ CHIRP-MS and AP-MS to map viral RNA-protein and protein-protein interactions, identifying 671 host factors involved in PDCoV infection. The work is well-designed, and provides valuable insights into coronavirus-host interactions with potential therapeutic implications.

Reviewer #2: This article focused on screening host-interacting proteins using PDCoV genomic RNA and encoded proteins as bait, revealing preferential differences between vRNA and vProtein in their interactions with host proteins during PDCoV infection. After functional clustering analysis of all host-interacting proteins, they constructed a protein interaction network between PDCoV and its host. Subsequently, SYNCRIP was selected as a key host protein to explore its functional mechanisms in viral infection and assess its potential as an antiviral target. Overall, I consider this an excellent and highly innovative study with substantial evidence. It provides valuable insights into how viruses hijack and reprogram host cells, and comparing interaction networks between different viruses and hosts holds positive significance for identifying broad-spectrum antiviral targets. I recommend minor revisions for acceptance.

**Part II – Major Issues: Key Experiments Required for Acceptance**

Reviewer #1: 1. While the interaction between SYNCRIP and N proteins is well-documented, the exact molecular mechanism by which SYNCRIP prevents HUWE1-mediated ubiquitination could be further explored. Structural studies or detailed mutagenesis analysis might strengthen this section.

2.The in vivo piglet study with Isoforsythiaside shows promise, but additional pharmacokinetic/pharmacodynamic data would strengthen the therapeutic potential claim.

3. The comparison with other coronavirus-host interactomes (Fig 2E) is somewhat superficial. A more detailed discussion of conserved vs. unique interaction patterns would be valuable.

4. Some quantitative proteomics data (e.g., spectral counts, peptide counts) are not fully reported, making it difficult to assess interaction confidence levels.

Reviewer #2: I recommend minor revisions for acceptance.

**Part III – Minor Issues: Editorial and Data Presentation Modifications**

Reviewer #1: 5. The discussion could better integrate findings with current reports on coronavirus-host interactions.

Reviewer #2: Specific questions regarding this study are as follows:

Introduction section舄

(1) Could you supplement the applications of CHIRP-MS and AP-MS in screening host-interacting proteins for other viruses?

(2) Are there other studies reporting the combined use of CHIRP-MS and AP-MS on the same pathogen?

(3) Could you elaborate on SYNCRIP's functions and its role in viral infection?

Results section舄

(1) Viral RNA forms secondary structures that may affect the complementary pairing of labeled probes. How did you improve the binding efficiency between viral RNA and labeled probes? Did you test their binding efficiency?

(2) Line 146: How did you distinguish viral RNA from host RNA?

(3) Coronavirus sgRNAs are expressed unevenly. Did you test the proportions of each sgRNA in the recovered viral RNA?

(4) Line 154: LC-MS detected signals from ORF1ab. What are the proportions of the nsps formed by ORF1ab cleavage?

(5) Lines 188-206: I am particularly interested in the interaction features of the host-interacting proteins associated with the viral key proteins nsp5, nsp12, and S protein. Could you supplement these here?

(6) Line 279: Did you use an interaction protein library from porcine-related viruses?

(7) Line 321: Could you provide the binding status of these six significantly changed host proteins during infection with vRNA or vProtein?

(8) Line 356: Did you examine SYNCRIP's expression levels after in vivo viral infection?

(9) Does PDCoV infection affect the localization of endogenous SYNCRIP?

(10) Line 373: Besides PEDV, did you test the replication of other porcine-related viruses in SYNCRIP-knockout cells?

(11) Fig 6A: Could you provide fluorescence localization signal images from confocal microscopy?

(12) Line 449: What was the concentration of B18622 in PK1 cells?

(13) Line 469: Please list the amino acid differences of SYNCRIP proteins across species in a table.

(14) Line 473: SYNCRIP interacts with the N protein of other coronaviruses. Can SYNCRIP similarly promote the replication of other coronaviruses?

(15) Line 622: Could you explain the reasons for the low overlap between the receptors identified by CRISPR/Cas9 screening and this study's database?

(16) Please standardize the spelling of "ChIRP-MS" throughout the manuscript.

PLOS authors have the option to publish the peer review history of their article (what does this mean? ). If published, this will include your full peer review and any attached files.

**Do you want your identity to be public for this peer review?** For information about this choice, including consent withdrawal, please see our Privacy Policy .

Reviewer #1: No

Reviewer #2: No

**Figure resubmission:**

**Reproducibility:**



---

## [Decision Letter · Decision Letter 1]

11 Oct 2025

Dear associate professor Yang,

We are pleased to inform you that your manuscript 'A comprehensive PDCoV-host proteome interaction map reveals potential antiviral targets' has been provisionally accepted for publication in PLOS Pathogens.

Best regards,

Luis Martínez-Sobrido

Academic Editor

PLOS Pathogens

Alexander Gorbalenya

Section Editor

PLOS Pathogens

Sumita Bhaduri-McIntosh

Editor-in-Chief

PLOS Pathogens

orcid.org/0000-0003-2946-9497

Michael Malim

Editor-in-Chief

PLOS Pathogens

orcid.org/0000-0002-7699-2064

Reviewer Comments (if any, and for reference):

Reviewer's Responses to Questions

**Part I - Summary**

Reviewer #1: (No Response)

Reviewer #2: (No Response)

**Part II – Major Issues: Key Experiments Required for Acceptance**

Reviewer #1: (No Response)

Reviewer #2: (No Response)

**Part III – Minor Issues: Editorial and Data Presentation Modifications**

Reviewer #1: (No Response)

Reviewer #2: (No Response)

PLOS authors have the option to publish the peer review history of their article (what does this mean? ). If published, this will include your full peer review and any attached files.

**Do you want your identity to be public for this peer review?** For information about this choice, including consent withdrawal, please see our Privacy Policy .

Reviewer #1: No

Reviewer #2: No

---

## [Editor Report · Acceptance letter]

Dear associate professor Yang,

We are delighted to inform you that your manuscript, "A comprehensive PDCoV-host proteome interaction map reveals potential antiviral targets," has been formally accepted for publication in PLOS Pathogens.

Best regards,

Sumita Bhaduri-McIntosh

Editor-in-Chief

PLOS Pathogens

orcid.org/0000-0003-2946-9497

Michael Malim

Editor-in-Chief

PLOS Pathogens

orcid.org/0000-0002-7699-2064